# Revisiting Discrete Soft Actor-Critic

## Abstract

We study the adaption of soft actor-critic (SAC) from continuous action space to discrete action space. We revisit vanilla SAC and provide an in-depth understanding of its Q value underestimation and performance instability issues when applied to discrete settings. We thereby propose entropy-penalty and double average Q-learning with Q-clip to address these issues. Extensive experiments on typical benchmarks with discrete action space, including Atari games and a large-scale MOBA game, show the efficacy of our proposed method. Our code is at: `https://github.com/revisiting-sac/Revisiting-Discrete-SAC.git`.

## 1 Introduction

In the conventional model-free reinforcement learning (RL) paradigm, an agent can be trained by learning an approximator of action-value (Q) function (Mnih et al., 2015; Bellemare et al., 2017). The class of actor-critic algorithms (Mnih et al., 2016; Fujimoto et al., 2018) evaluates the policy function by approximating the value function. Motivated by maximum-entropy RL(Ziebart et al., 2008; Rawlik et al., 2012; Abdolmaleki et al., 2018), soft actor-critic (SAC) (Haarnoja et al., 2018a) introduces action entropy in the framework of actor-critic to achieve exploit-explore trade-off and it has achieved remarkable performance in a range of environments with continuous action spaces (Haarnoja et al., 2018b), and is considered as the state-of-the-art algorithm for domains with continuous action space, e.g., Mujoco (Todorov et al., 2012).

However, while SAC solves problems with continuous action space, it cannot be straight-forwardly applied to discrete domains since it relies on the reparameterization of Gaussian policies to sample actions, in which the action in discrete domains is categorical. Soft-DQN (Vieillard et al., 2020) provides a simple way to discretize SAC by adopting the maximum-entropy RL to DQN (Mnih et al., 2013). However, Soft-DQN utilizes only a Q-value parametrization to bypass the policy parameterization. Another discretization of the continuous action output and Q value in vanilla SAC is an obvious strategy suggested by (Christodoulou, 2019) to adapt SAC to discrete domains, resulting in the discrete version of SAC, denoted as discrete SAC (DSAC) throughout the paper. However, it is counter-intuitive that the empirical experiments in subsequent efforts (Xu et al., 2021) indicate that discrete SAC performs poorly in discrete domains, e.g., Atari games. We believe that the idea of maximum entropy RL is applicable to both discrete and continuous domains. However, so far, extending the maximum-entropy based SAC algorithm to discrete domains still lacks a commonly accepted practice in the community. Therefore, in this paper, similar to the motivation of DDPG (deep deterministic policy gradient) (Lillicrap et al., 2016) which adapts DQN (deep Q networks) (Mnih et al., 2013) from discrete action space to continuous action space, we aim to optimize SAC for discrete domains.

Previous studies (Xu et al., 2021; Wang & Ni, 2020) have analyzed the reasons for the performance disparity of SAC between continuous and discrete domains. Reviewing from the perspective of automating entropy adjustment, an unreasonable setting of target-entropy for temperature $\alpha$ may break the SAC value–entropy trade-off (Wang & Ni, 2020; Xu et al., 2021). Furthermore, the function approximation errors of Q-value are known to lead to estimation bias and hurt performance in actor-critic methods (Fujimoto et al., 2018). To avoid overestimation bias, both discrete SAC and continuous SAC resort to clipped double Q-learning (Fujimoto et al., 2018) for actor-critic algorithms. On the contrary, using the lower bound approximation to the critic can lead to underestimation bias, which makes the policy fall into pessimistic underexplore, as pointed by (Ciosek et al., 2019; Pan et al., 2020), particularly when the reward is sparse. However, existing

works only focus on continuous domains (Ciosek et al., 2019; Pan et al., 2020), while SAC for discrete cases remains to be less explored.

In addition to the aforementioned issues, we conjecture that the reason discrete SAC fails also includes the absence of policy update constraints. Intuitively, a sudden change in the policy causes a shift in the entropy, which generates a rapidly changing target for the critic network, due to the soft Q-learning objective. Meanwhile, the critic network in SAC needs time to adapt to the oscillating target process and exacerbates policy instability.

To address the above challenges, in this paper, we first design test cases to replicate the failure modes of vanilla discrete SAC, exposing its inherent weaknesses regarding training instability and Q-value underestimation. Then, accordingly, to stabilize the training, we develop entropy-penalty on the policy optimization objective to constrain policy update; and to confine the Q value within a reasonable range, we develop double average Q-learning with Q-clip. We use Atari games (the default testbed for RL algorithm for discrete action space) to verify the effectiveness of our optimizations. We also deploy our method to the Honor of Kings 1v1 game, a large-scale MOBA game used extensively in recent RL advances (Ye et al., 2020b;c;a; Wei et al., 2022), to demonstrate the scale-up capacity of our optimized discrete SAC.

To sum up, our contributions are:

- We pinpoint two failure modes of discrete SAC, regarding policy instability and underestimated Q values, respectively.

- To alleviate policy instability, we propose entropy-penalty to constrain the policy update in discrete SAC.

- To deal with the underestimation bias of Q value in discrete SAC, we propose double average Q-learning with Q-clip to estimate the state-action value.

- Extensive experiments on Atari games and a large-scale MOBA game show the superiority of our method.

## 2 Related Work

We review recent efforts on algorithmic improvements to soft actor-critic.

**Adaption of Action Space**. The most relevant works to this paper are: vanilla discrete SAC (Christodoulou, 2019) , TES-SAC (Xu et al., 2021) and Soft-DQN (Vieillard et al., 2020). Discrete SAC replace the Gaussian policy with a categorical policy and discretize the Q-value output to adapt SAC from continuous action space to discrete action space. However, as we will point out, a direct discretization of SAC will have certain failure modes with poor performance. TES-SAC point out that it is counter-intuitively that SAC does not work well for discrete action space. They propose a new scheduling method for the target entropy parameters in discrete SAC. However, they contend that the failure modes of discrete SAC are due to unreasonable target-entropy parameters, whereas, we point out that poor adaption of discrete SAC results from policy instability and underestimated Q value. Soft-DQN has discretized SAC by adopting the maximum-entropy RL to DQN. Unlike SAC, which uses both policy and Q value parameters simultaneously, Soft-DQN utilizes only a Q value parametrization and directly applies a softmax operation to the Q-values to take actions.

**Q Estimation**. Previous works (Fujimoto et al., 2018; Ciosek et al., 2019; Pan et al., 2020; Duan et al., 2021) have already expressed concerns about the estimation bias of Q value for SAC. SD3 (Pan et al., 2020) proposes to reduce the Kurtosis distribution of Q approximate by using the softmax operator on the original Q value output to reduce the overestimation bias. OAC (Ciosek et al., 2019) constrains the Q value approximation objective by calculating the upper and lower boundaries of two Q-networks. DSAC (Duan et al., 2021) replaces the Q learning target with the expected reward sum obtained from the current state to the end of the episode and uses multi-frame estimates target to reduce overestimation. The methods unavoidably increase the cost of complexity while obtaining an accurate Q-value overestimation with continuous action

spaces. However, little research is on discrete settings. By comparison, we propose an approximation method by replacing double average Q outputs to be the learning target and clipping the current Q value with the target network. We prevent both overestimation and underestimation with little extra computational cost. Maxmin Q-learning (Lan et al., 2020) controls estimation bias by minimizing over the full ensemble in the target. MME (Han & Sung, 2021) extends max-min operation to the entropy framework to adapt to SAC. REM (Agarwal et al., 2020) ensembles Q-value estimations with a random convex combination to enhance generalization in the offline setting. REDQ (Chen et al., 2021b) reduces the estimation bias by minimizing a random subset of Q-functions. AEQ (Gong et al., 2023) adjusts the estimation bias by using the mean of Q-functions minus their standard deviation. Compared to these methods that employ ensemble multiple Q-estimators, our approach focuses on reducing underestimation bias for double Q-estimators to enhance exploration.

**Performance Stability**. Flow-SAC (Ward et al., 2019) applies a technique called normalizing flows policy on continuous SAC leading to finer transformation that improves training stability when exploring complex states. However, applying normalizing flows to discrete domains will cause a degeneracy problem (Horvat & Pfister, 2021), making it difficult to transfer to discrete actions. SAC-AWMP (Hou et al., 2020) improves the stability of final policy by using weighted mixture to combine multiple policies. The cost, based on this method, is that network parameters and inference speed are significantly increased. ISAC (Banerjee et al., 2022) increases SAC stability by mixing prioritized learning samples and on-policy samples, which essentially enables the actor to repeat learns states with drastic changing. Repeatedly learning priority samples, however, runs the risk of settling into a local optimum. By comparison, our method improves the stability of policy in case of drastic state changes with an entropy constraint.

## 3 Preliminaries

In this section, we provide a brief overview of the symbol definitions of SAC for discrete action space.

Follow by the maximum entropy framework, SAC adds an entropy term $\mathbb{H}(\pi(\cdot \mid s))$ , as a regularization term to the policy gradient objective:

$$\pi^* = \underset{\pi}{\operatorname{argmax}} \sum_{t=0}^{T} \underset{\substack{s_t \sim p \\ a_t \sim \pi}}{\mathbb{E}}[r(s_t, a_t) + \alpha \mathbb{H}(\pi(\cdot \mid s))], \tag{1}$$

where

$$\begin{aligned} \mathbb{H}(\pi(\cdot \mid s)) &= -\int \pi(a \mid s) \log \pi(a \mid s) \mathrm{d}a \\ &= \underset{a \sim \pi(\cdot \mid s)}{\mathbb{E}}[-\log \pi(a \mid s)] \end{aligned} \tag{2}$$

where $\pi$ is a policy, $\pi^*$ is the optimal policy, and $\alpha$ is the temperature parameter that determines the relative importance of the entropy term versus the reward $r$, thus controls the stochasticity of the optimal policy.

**Soft Bellman Backup** The soft Q-function, parametrized by $\theta$, is updated by reducing the soft bellman error as described in the next subsection:

$$J_Q(\theta) = \frac{1}{2} \left( r(s_t, a_t) + \gamma V(s_{t+1}) - Q_\theta(s_t, a_t) \right)^2, \tag{3}$$

where $V(s_t)$ defines the soft state value function, which represents the expected reward estimate that policy obtains from the current state to the end of the trajectory.

$$V(s_t) = \mathbb{E}_{a_t \sim \pi}[Q_\theta(s_t, a_t) - \alpha \log(\pi(a_t \mid s_t))]. \tag{4}$$

Soft actor-critic minimizes soft Q-function with final soft bellman error:

$$J_Q(\theta) = \mathbb{E}_{(s_t, a_t) \sim D}[\frac{1}{2}(Q_\theta(s_t, a_t) - (r(s_t, a_t) + \gamma \mathbb{E}_{s_{t+1} \sim p(\cdot \mid s_t, a_t)}[V(s_{t+1})]))^2], \tag{5}$$

where D is a replay buffer replenished by rollouts of the policy $\pi$ interacting with the environment. In the implementation, SAC (Haarnoja et al., 2018a) uses the minimum of two delayed-update target-critic network outputs as the soft bellman learning objective to reduce overestimation. The formula is expressed as

$$V(s_{t+1}) = \min_{i=1,2} \mathbb{E}_{a_t \sim \pi}[Q_{\theta_i'}(s_{t+1}, a_{t+1}) - \alpha \log(\pi(a_{t+1} \mid s_{t+1}))], \tag{6}$$

where $Q_{\theta_i'}$ represents $i$-th target-critic network.

**Policy Update Iteration** The policy, parameterized by $\phi$, is a distillation of the softmax policy induced by the soft Q-function. The discrete SAC policy directly maximizes the probability of discrete actions, in contrast to the continuous SAC policy which optimizes the two parameters of the Gaussian distribution. Then the discrete SAC policy is updated by minimizing KL-divergence between the policy distribution and the soft Q-function.

$$\pi_{\phi_{new}} = \underset{\pi_{\phi_{old}} \in \Pi}{\operatorname{argmin}} D_{\mathrm{KL}}(\pi_{\phi_{old}}(. \mid s_t) \| \frac{\exp(\frac{1}{\alpha} Q^{\pi_{\phi_{old}}}(s_t, .))}{Z^{\pi_{\phi_{old}}}(s_t)}). \tag{7}$$

Note that the partition function $Z^{\pi_{\phi_{old}}}(s_t)$ is a normalization term that can be ignored since it does not affect the gradient with respect to the new policy. The resulting optimization objective of the policy is as followed:

$$J_\pi(\phi) = \mathbb{E}_{s_t \sim D}[\mathbb{E}_{a_t \sim \pi_\phi}[\alpha \log(\pi_\phi(a_t \mid s_t)) - Q_\theta(s_t, a_t)]]. \tag{8}$$

**Automating Entropy Adjustment** The entropy parameter temperature $\alpha$ regulates the value-entropy balance in soft Q learning. The SAC paper proposes using the temperature Lagrange term to automatically tune the temperature $\alpha$. The following equation can be regarded as the optimization objective satisfying an entropy constraint.

$$\max_{\pi_{0:T}} \mathbb{E}_{\rho_\pi}[\sum_{t=0}^{T} r(\mathbf{s}_t, \mathbf{a}_t)] \tag{9}$$

$$\text{s.t. } \mathbb{E}_{(s_t, a_t) \sim \rho_\pi}[-\log(\pi_t(\mathbf{a}_t \mid \mathbf{s}_t))] \geq \mathcal{H}, \forall t,$$

where $\mathcal{H}$ is the desired minimum expected entropy. Optimizing the Lagrangian term $\alpha$ involves minimizing:

$$J(\alpha) = \mathbb{E}_{(a|s) \sim \pi_t}[-\alpha \log \pi_t(a_t \mid s_t) - \alpha \mathcal{H}]. \tag{10}$$

By setting a loose upper limit on the target entropy $\mathcal{H}$, SAC achieves automatic adjustment of temperature $\alpha$. Typically, the target entropy is set to $0.98 * -log(\frac{1}{dim(Actions)})$ for discrete(Christodoulou, 2019) and $-dim(Actions)$ for continuous actions(Haarnoja et al., 2018b).

## 4 Failure Modes of Vanilla Discrete SAC

We start by outlining the failure modes of the vanilla discrete SAC and then analyze under what circumstances the standard choices of vanilla discrete SAC perform poorly.

### 4.1 Drastic Changes of Policy

The first failure mode comes from a scenario where policy and Q-learning fail to recover from an erratic training process when the state distribution suddenly changes. The maximum entropy mechanism in SAC effectively balances exploration and exploitation. However, due to the existence of entropy term in the soft bellman error, the policy update iteration (Eq. 8) is strongly coupled with the Q-learning iteration (Eq. 5). This learning paradigm poses a particular risk that the abrupt change of state distribution could lead to policy instability and entropy chattering, consequently, the Q learning target becomes unstable, which can in turn deteriorate the policy learning. To illustrate this issue more concretely, we take the following Atari game as an example.

Consider the training process of discrete SAC in the Atari game Asterix, as shown in Fig. 1. At the early training stage, policy tends to explore randomly in the environment, which accelerates the change of state

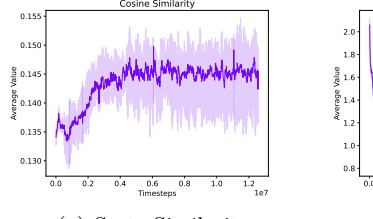 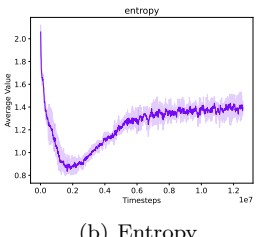 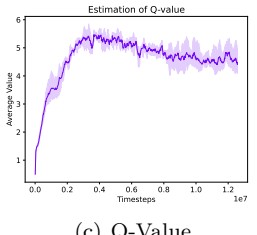 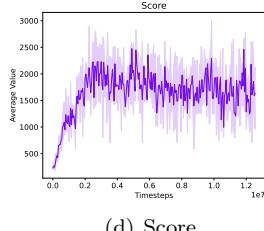

(a) State Similarity      (b) Entropy      (c) Q-Value      (d) Score

Figure 1: Measuring cosine similarity of states, policy action entropy, estimation of Q-value and score on Atari Game Asterix environment with discrete SAC over 10 million time steps.

distribution. As shown in Fig. 1(a), we measure the cosine distance between state distributions induced by adjacent policies (i.e., $\pi_k$ and $\pi_{k+10}$), and find that the change of state distribution is becoming more and more drastic during training. As the learning process goes on, the policy entropy drops rapidly and the action probabilities become deterministic quickly (Fig. 1(b)). At the same time, the drastic change of policy entropy misleads the learning process of policy and Q-value, and thus both Q-value and policy fall into local optimum (Fig. 1(b) and Fig. 1(c)). Since both policy and Q-value converge to local optimum, it becomes hard for the policy to explore efficiently in the later training stage. Even the policy entropy re-rises in the later stage (Fig. 1(b))), the performance of policy does not improve anymore (Fig. 1(d)).

To better understand why this undesirable behavior occurs, we inspect the gradient of the soft bellman object calculated by formula 5.

$$\hat{\nabla}_\theta J_Q(\theta) = \nabla_\theta Q_\theta(\mathbf{a}_t, \mathbf{s}_t)(Q_\theta(\mathbf{s}_t, \mathbf{a}_t) - (r(\mathbf{s}_t, \mathbf{a}_t) + \gamma(Q_\theta(\mathbf{s}_{t+1}, \mathbf{a}_{t+1}) - \alpha \log(\pi_\phi(\mathbf{a}_{t+1} \mid \mathbf{s}_{t+1}))))). \tag{11}$$

As shown in Eq. 11, the improvement of $Q_\theta(\mathbf{s}_t, \mathbf{a}_t)$ relies on the Q-estimation of next states and policy entropy. However, the drastically shifting entropy can increase the uncertainty of gradient updates and mislead the learning of Q-network. Since the policy is induced by the soft Q-network, the policy can also become misleading and hurt performance. To mitigate this phenomenon, the key is to ensure the smoothness of policy change so as to maintain stable training. We will introduce in the next section how to constrain the randomness of the policy to ensure smooth changes in policy.

## 4.2 Pessimistic Exploration

The second failure mode comes from pessimistic exploration due to the double Q-learning mechanism. The double Q-learning trick has been widely used in value-based or actor-critic RL algorithms for both discrete (e.g., Double DQN (Van Hasselt et al., 2016)) and continuous (e.g., SAC (Haarnoja et al., 2018a)) domains. In discrete domains, due to the max operator, DQN tends to suffer from overestimation bias. Double DQN uses the double Q-learning trick to mitigate this issue. In continuous domains, inspired by Double DQN, TD3 (Fujimoto et al., 2018) and SAC adopt clipped double Q-learning to mitigate overestimation. Empirical results demonstrate that the clipped double Q-learning trick can boost the performance of SAC in continuous domains, but the impact of this trick has remained to be unclear in discrete domains. Therefore, we need to revisit the use of clipped double Q-learning for discrete SAC.

In our experiments, in discrete domains, we find that discrete SAC tends to suffer from underestimation bias instead of overestimation bias. This underestimation bias can cause pessimistic exploration, especially in the case of sparse reward. Here we give an illustration of how the popularly used clipped double Q-learning trick causes the issue of underestimation bias and how the policy used this trick tends to converge to suboptimal actions for discrete action spaces. Our work complements previous work with a more in-depth analysis of clipped double Q-learning. We demonstrate the existence of underestimation bias and then illustrate the impact of underestimation on Atari games.

To analyze the estimated bias $\epsilon$, we introduce the mathematical expression of the soft state-value function:

$$V(s_t) = \mathbb{E}_{a_t \sim \pi}[Q(s_t, a_t) - \alpha \log(\pi(a_t \mid s_t))], \tag{12}$$

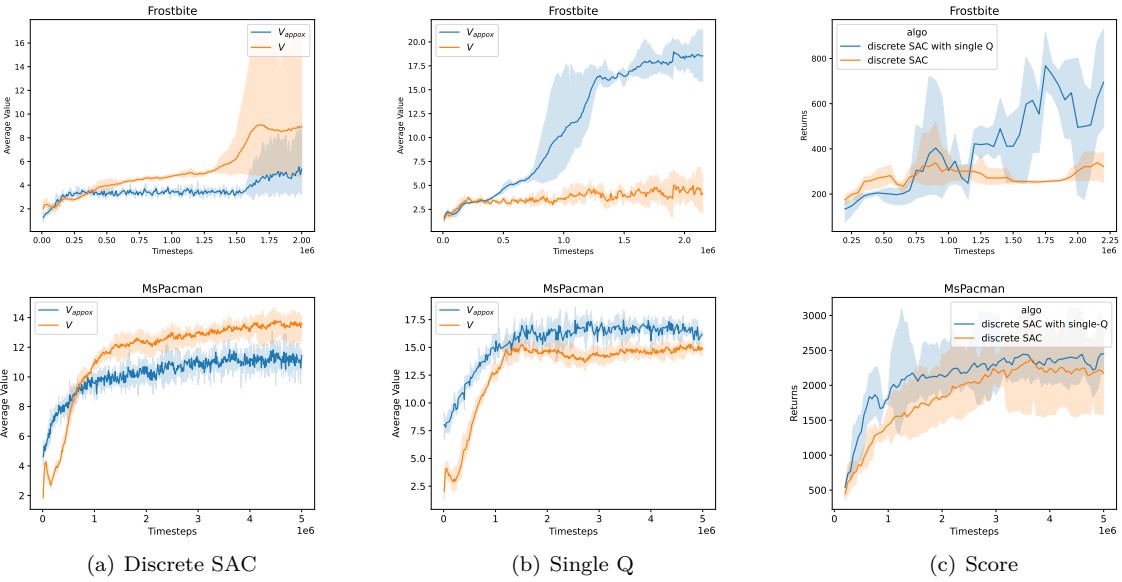

(a) Discrete SAC          (b) Single Q          (c) Score

Figure 2: The results of Atari game Frostbite/MsPacman environment over 2/5 million time steps: a) Measuring Q-value estimates of discrete SAC; b) Measuring Q-value estimates of discrete SAC with single Q; c) Score comparison between discrete SAC and discrete SAC with single Q.

where $Q(s_t, a_t)$ represents the true Q-value. In practice, SAC uses the clipped double Q-learning trick. The learning target of soft state-value function can be written as:

$$V_{appox}(s_t) = \mathbb{E}_{a_t \sim \pi} \min_{i=1,2} [Q_{\theta'_i}(s_t, a_t) - \alpha \log(\pi(a_t \mid s_t))], \tag{13}$$

where $Q_{\theta'}$ represents estimation of target-critic networks parameterized by $\theta'$. The estimated bias for $Q'_{\theta_i}$ can be calculated as $\epsilon_i = Q_{\theta'_i}(s, a) - Q(s, a)$. On the one hand, when $\epsilon_1 > \epsilon_2 > 0$, using the clipped double Q-learning trick can help mitigate overestimation error due to the *min* operation. On the other hand, when $\epsilon_1 < \epsilon_2 < 0$ or $\epsilon_1 < 0 < \epsilon_2$, the clipped double Q-learning trick will lead to underestimation (i.e., $V_{appox} < V$) and consequently result in pessimistic exploration (Pan et al., 2020; Ciosek et al., 2019).

Does this theoretical underestimate occur in practice for discrete SAC and hurt the performance? We answer this question by showing the influence of the clipped double Q-learning trick for discrete SAC in Atari games, as shown in Fig. 2. Here we show a comparison between the true value and the estimated value. The results are averaged over 3 independent experiments with different random seeds. We find that, in Fig. 2(a), the approximate values are lower than the true value over time, which demonstrates the issue of underestimation bias. At the same time, we also run experiments for discrete SAC with single Q (DSAC-S), which uses a single Q-value for bootstrapping instead of clipped double Q-values. As shown in Fig. 2(b), without the clipped double Q-learning trick, the estimated value of DSAC-S is higher than the true value and thus has overestimation bias. However, in Fig. 2(c), we discover that even though DSAC-S suffers from overestimation bias, it performs much better than discrete SAC which adopts the clipped double Q-learning mechanism. This indicates that the clipped double Q-learning trick can lead to pessimistic exploration issues and hurt the agent's performance.

## 5 Improvements of SAC Failure Modes

We provide two simple alternatives, which are the surrogate objective with entropy-penalty and double average Q-learning with Q-clip, so as to avoid the two failure modes of discrete SAC discussed in Section 4. Combining these two modifications, we propose a new algorithm, called discrete-SAC with entropy-penalty and double average Q-learning with Q-clip. The pseudo code is provided in Algorithm 1.

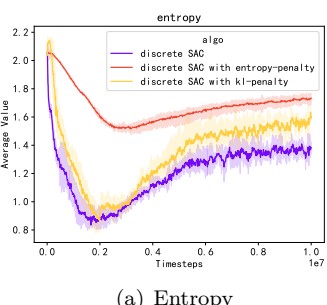
(a) Entropy

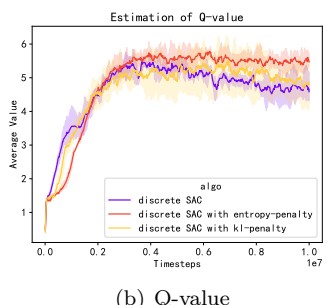
(b) Q-value

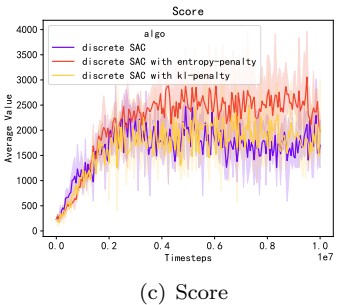
(c) Score

Figure 3: Measuring policy action entropy, estimation of Q-value, and score on Atari game Asterix compared between discrete SAC, discrete SAC with KL-penalty and discrete SAC with entropy-penalty over 10 million time steps.

### 5.1 Entropy-Penalty

The drastic change of entropy affects the optimization of the Q-value. Simply removing the entropy term will injure the exploration ability under the framework of maximum entropy RL. An intuitive solution is to introduce an entropy penalty in the objective of policy to avoid entropy chattering. We will introduce how to incorporate the entropy penalty in the learning process for the discrete SAC algorithm.

Recall the objective of policy in SAC as in Eq. 8. For a mini-batch transition data pair $(s_t, a_t, r_r, s_{t+1})$ sampled from the replay buffer, we add an extra entropy term $\mathbb{H}\pi_{old}$ to the transition tuple which reflects the randomness of policy (i.e., $(s_t, a_t, r, s_{t+1}, \mathbb{H}\pi_{old})$), where $\pi_{old}$ denotes the policy used for data sampling. We calculate the entropy penalty by measuring the distance between $\mathbb{H}\pi_{old}$ and $\mathbb{H}\pi$. Formally, the objective of the policy is as the following:

$$
\begin{aligned}
J_\pi(\phi) = {} & \mathbb{E}_{s_t \sim D}[\mathbb{E}_{a_t \sim \pi_\phi}[\alpha \log(\pi_\phi(a_t \mid s_t)) - Q_\theta(s_t, a_t)]] \\
& + \beta \cdot \frac{1}{2}\mathbb{E}_{s_t \sim D}([\mathbb{E}_{a_t \sim \pi_{\phi_{old}}}[-\log(\pi_{\phi_{old}})] - \mathbb{E}_{a_t \sim \pi_\phi}[-\log(\pi_\phi)])^2,
\end{aligned}
\tag{14}
$$

where $\mathbb{E}_{a_t \sim \pi_{\phi_{old}}}[-\log(\pi_{\phi_{old}})]$ represents policy entropy of $\pi_{\phi_{old}}$, $\mathbb{E}_{a_t \sim \pi_\phi}[-\log(\pi_\phi)]$ represents policy entropy of $\pi_\phi$, and $\beta$ denotes a coefficient for the penalty term and is set to 0.5 in this paper. By constraining the policy objective with this penalty term, we increase the stability of the learning process of policy.

Fig. 3 shows the training curves to demonstrate how the entropy penalty mitigates the failure mode of policy drastic change. In Fig. 3(a), the entropy of discrete SAC (the purple curve) drops quickly and the policy falls into a local optimum at the early training stage. Later, the policy stops improving and even suffers from performance deterioration as shown in the purple curves in Fig. 3(b) and Fig. 3(c). On the contrary, our proposed method (i.e., discrete SAC with entropy-penalty) demonstrates better stability than discrete SAC. As shown in Fig. 3(a), with entropy penalty, the policy changes smoothly during training. Consequently, compared with discrete SAC, the policy in our approach can keep improving during the whole training stage and does not suffer from performance drop at the later training stage (the red curves in Fig. 3(b) and Fig. 3(c)). KL penalty (the yellow curve) can mitigate the drastic changes of policy, preventing training collapse. However, the effectiveness of the KL penalty in reducing drastic changes is inferior to the entropy penalty. The final Q-value and score are lower than those with the entropy penalty, with a difference of 12% and 23% respectively.

The entropy-penalty term $\frac{1}{2}\mathbb{E}_{s_t \sim D}([\mathbb{E}_{a_t \sim \pi_{\phi_{old}}}[-\log(\pi_{\phi_{old}})] - \mathbb{E}_{a_t \sim \pi_\phi}[-\log(\pi_\phi)])^2$, in conjunction with the temperature $\alpha$, jointly regulates the exploration of policy. Different from other trust region methods such as KL constraint (Schulman et al., 2015) or clipping surrogate objective (Schulman et al., 2017), our method penalizes the change of action entropy between old and new policies to address policy instability during training. By adding regularization in entropy space instead of policy space, out method can mitigate the drastic changes of policy entropy while maintaining the inherent exploratory ability of SAC (as shown in Fig. 3(a), the policy entropy changes smoothly and keeps at a relatively high value to encourage exploration).

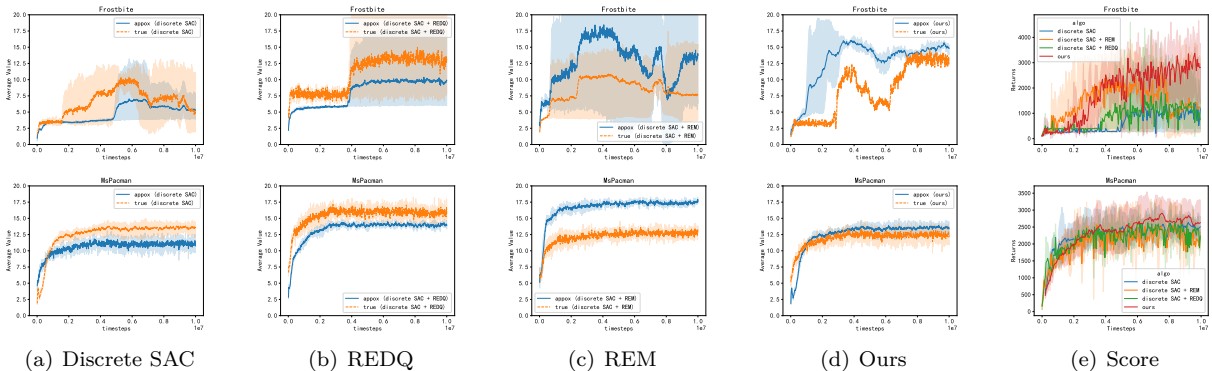

(a) Discrete SAC     (b) REDQ     (c) REM     (d) Ours     (e) Score

Figure 4: Measuring estimation of Q-value and score on Atari Game Frostbite/MsPacman environment compared between discrete SAC, discrete SAC with REDQ, discrete SAC with REM and ours (discrete SAC with double average Q-learning with Q-clip) over 10 million steps.

## 5.2 Double average Q-learning with Q-clip

While several approaches(Ciosek et al., 2019; Pan et al., 2020) have been proposed to reduce underestimation bias, they are not straightforward to be applied to discrete SAC due to the use of Gaussian distribution. In this section, we introduce a novel variant of double Q-learning to mitigate the underestimation bias for discrete SAC.

In practice, discrete SAC uses clipped double q-learning with a pair of target critics $(Q_{\theta'_1}, Q_{\theta'_2})$, and the learning target of these two critics is:

$$y = r + \gamma \min_{i=1,2} Q_{\theta'_i}(s', \pi(s')). \tag{15}$$

When the Q-function is approximated by neural networks, there exists unavoidable bias in the critics. Since policy is optimized with respect to the low bound of double critics, for some states, we will have $Q_{\theta'_2}(s, \pi_\phi(s)) > Q_{true} > Q_{\theta'_1}(s, \pi_\phi(s))$. This is problematic because $Q_{\theta'_1}(s, \pi_\phi(s))$ will generally underestimate the true value, and this underestimated bias will be further exaggerated during the whole training phase, which results in pessimistic exploration.

To address this problem, we propose to mitigate the underestimation bias by replacing the *min* operator with *avg* operator. This results in taking the average between the two estimates, which we refer to as *double average Q-learning*:

$$y = r + \gamma \cdot \text{avg}(Q_{\theta'_1}(s', \pi(s')), Q_{\theta'_2}(s', \pi(s'))). \tag{16}$$

By doing so, the underestimated bias of the lower bound of double critics can be mitigated by the other critic. To improve the stability of the Q-learning process, inspired by value clipping in PPO (Schulman et al., 2017), we further add a clip operator on the bellman error to prevent drastic updates of Q-network. The modified bellman loss of Q-network is as following:

$$\mathcal{L}(\theta_i) = \max\left((Q_{\theta_i} - y)^2, (Q_{\theta'_i} + \text{clip}(Q_{\theta_i} - Q_{\theta'_i}, -c, c)) - y)^2\right), \tag{17}$$

where $Q_{\theta_i}$ represents the critic network's estimate, $Q_{\theta i'}$ represents estimation of target-critic networks, and $c$ is the hyperparameter denoting the clip range. This clipping operator prevents the Q-network from performing incentive update that goes beyond the clip range. In this way, the Q-learning process is more robust to the abrupt change of data distribution. Combining the clipping mechanism (Eq. 17) with double average Q-learning (Eq. 16), we refer our proposed approach as *double average Q-learning with Q-clip*.

Fig. 4 demonstrates the effectiveness of our approach. We compare the discrete SAC and various ensemble Q-estimation methods, including Randomized Ensembled Double Q-learning (REDQ) Chen et al. (2021b)

Table 1: Mean and median normalized scores of discrete SAC , TES-SAC , Rainbow , Soft-DQN and our method across all 20 Atari games at 1M and 10M steps.

| | Discrete SAC(1M) | TES-SAC(1M) | Soft-DQN(1M) | Ours(1M) | Rainbow(10M) | Discrete SAC(10M) | Soft-DQN(10M) | Ours(10M) |
|---|---|---|---|---|---|---|---|---|
| Mean | 0.5% | 3.0% | **41.7**% | 38.5% | 187.4 % | 151.4% | 199.2% | **220.0**% |
| Median | 0.4% | 2.1% | **20.0**% | 11.1% | 79.2 % | 90.8% | 107.7% | **114.1**% |

and Random Ensemble Mixture (REM) Agarwal et al. (2020), with our proposed method, Double Average Q-learning with Q-clip. In Fig. 4(a), the Q-value estimate of discrete SAC is underestimated than the true value, therefore, the policy of discrete SAC suffers from pessimistic exploration and results in poor performance (blue curve in Fig. 4(e)). On the contrary, in Fig. 4(d), with double average Q-learning and Q-clip, the Q-value estimate gets rid of underestimation bias and improves quickly at the early training stage. The improvement of Q-value carries over to the performance of policy, consequently, our approach outperforms baseline discrete SAC by a large margin (Fig. 4(e)). The result also demonstrates that even though REDQ has less estimation bias in Fig. 4(b), it still suffers from underestimation bias, leading to suboptimal performance due to pessimistic exploration. Although REM addresses the underestimation issue in Fig. 4(c), the overestimation bias of REM significantly exceeds that of our proposed method, resulting in a rapid decline in performance at 8 million steps. In Fig. 4(d), we also notice that the Q-value overestimates the true value during the early training stage but finally converges to the true value after finishing the whole training process. This encourages early exploration, which is consistent to the principle of optimism in the face of uncertainty (Kearns & Singh, 2002).

## 6 Experiments

In this section, we analyze the main experimental results. First, we compare our improved SAC with the three most related works, i.e., discrete SAC (Christodoulou, 2019) , TES-SAC (Xu et al., 2021) , Soft-DQN (Vieillard et al., 2020) and Rainbow(Hessel et al., 2018) which is widely accepted algorithm in the discrete domain. Then we perform ablation studies, comparing several SAC variants with entropy-penalty and double average Q-learning with Q-clip. Finally, we visualize the loss surfaces of our SAC to help understand the stability of the training process.

### 6.1 Experimental Setup

To evaluate our algorithm, we measure its performance in 20 Atari games which were chosen as the same as Christodoulou at al (Christodoulou, 2019) for a fair comparison. After a policy is trained for every 50000 steps, its performance is immediately evaluated by running the corresponding deterministic policy for 10 episodes. We execute 3 random seeds for each algorithm for a total of 10 million environment steps (or 40 million frames). For the baseline implementation of discrete-SAC, we use Tianshou [1]. We find that Tianshou's implementation performs better than the original paper by Christodoulou (Christodoulou, 2019). We use the default hyperparameters in Tianshou, and the hyperparameters are consistent across all 20 games.

We start the game with up to 30 no-op actions, similar to (Mnih et al., 2013), to provide the agent a random starting position. To obtain summary statistics across games, following Hasselt (Van Hasselt et al., 2016), we normalize the score for each game as follows:

$$\text{Score}_{\text{normalized}} = \frac{\text{Score}_{\text{agent}} - \text{Score}_{\text{random}}}{\text{Score}_{\text{human}} - \text{Score}_{\text{random}}}. \tag{18}$$

### 6.2 Overall Performance

Table 1 provides an overview of results. Detailed results are presented in the table 2 and Fig. 9. Note that TES-SAC is not open-sourced, we re-implement the algorithm according to the paper, but the performance is lower than the results reported in their paper. Hence we choose to trust the authors by using the normalized

---

[1]https://github.com/thu-ml/tianshou

Table 2: Raw scores across all 20 Atari games. For methods discrete SAC (1M) and TES-SAC(1M), the scores come from the corresponding paper, and the NE means the score does not exists in the original paper.

| Game | Discrete SAC (1M) | TES-SAC(1M) | Soft-DQN(1M) | Ours(1M) | Rainbow(10M) | Discrete SAC (10M) | Soft-DQN(10M) | Ours (10M) |
|---|---|---|---|---|---|---|---|---|
| Alien | 216.90 | 685.93 | 726.33 | **981.67** | 1798.33 | **2717.67** | 2018.00 | 2158.33 |
| Amidar | 7.9 | 42.07 | 130.03 | **132.97** | 394.23 | 354.77 | **438.80** | 407.20 |
| Assault | 350.0 | 337.03 | 881.97 | **1664.77** | 1802.53 | 7189.97 | **7258.87** | 6785.60 |
| Asterix | 272.0 | 378.5 | 676.67 | **733.33** | 5853.33 | 2860.00 | 3761.67 | **5993.33** |
| BattleZone | 4386.7 | 5790 | **7933.33** | 6266.67 | 24266.67 | 16850.00 | **24733.33** | 9466.67 |
| BeamRider | 432.1 | NE | 3321.60 | **3468.60** | 3310.40 | 7169.60 | 7048.20 | **10506.60** |
| Breakout | 0.7 | 2.65 | **46.17** | 11.47 | **492.93** | 29.03 | 155.83 | 60.43 |
| CrazyClimber | 3668.7 | 4.0 | **25390.00** | 20753.33 | 30286.67 | 126320.00 | 95156.67 | **140726.67** |
| Enduro | 0.8 | NE | **54.23** | 0.93 | 1517.70 | 1326.77 | 1144.07 | **2246.40** |
| Freeway | 4.4 | 13.57 | 17.70 | **20.17** | 20.13 | 15.73 | **32.30** | 20.17 |
| Frostbite | 59.4 | 81.03 | 294.33 | **347.00** | 4163.67 | 646.33 | 2959.00 | **4806.00** |
| Jamesbond | 68.3 | 31.33 | 273.33 | **368.33** | 656.67 | 1386.67 | 965.00 | **2085.00** |
| Kangaroo | 29.3 | **307.33** | 160.00 | 120.00 | 3716.67 | 2426.67 | 2703.33 | **5556.67** |
| MsPacman | 690.9 | 1408 | 1528.00 | **1639.00** | 2738.67 | **3221.33** | 2386.33 | 3175.67 |
| Pong | -20.98 | -20.84 | **20.00** | 15.53 | **20.93** | 20.37 | 20.73 | 20.37 |
| Qbert | 280.5 | 74.93 | 1400.83 | 986.67 | 15299.17 | 12946.67 | 14293.33 | **15325.83** |
| RoadRunner | 305.3 | NE | 5510.00 | **12793.33** | **45173.33** | 34043.33 | 33370.00 | 43203.33 |
| SpaceInvaders | 160.8 | NE | **488.83** | 383.50 | **1330.50** | 458.83 | 816.00 | 586.50 |
| Seaquest | 211.6 | 116.73 | 681.33 | **744.00** | 2105.33 | 1853.33 | **3438.67** | 2764.00 |
| UpNDown | 250.7 | 207.6 | **8727.33** | 8114.67 | 9110.00 | 17803.33 | **79313.00** | 63441.33 |

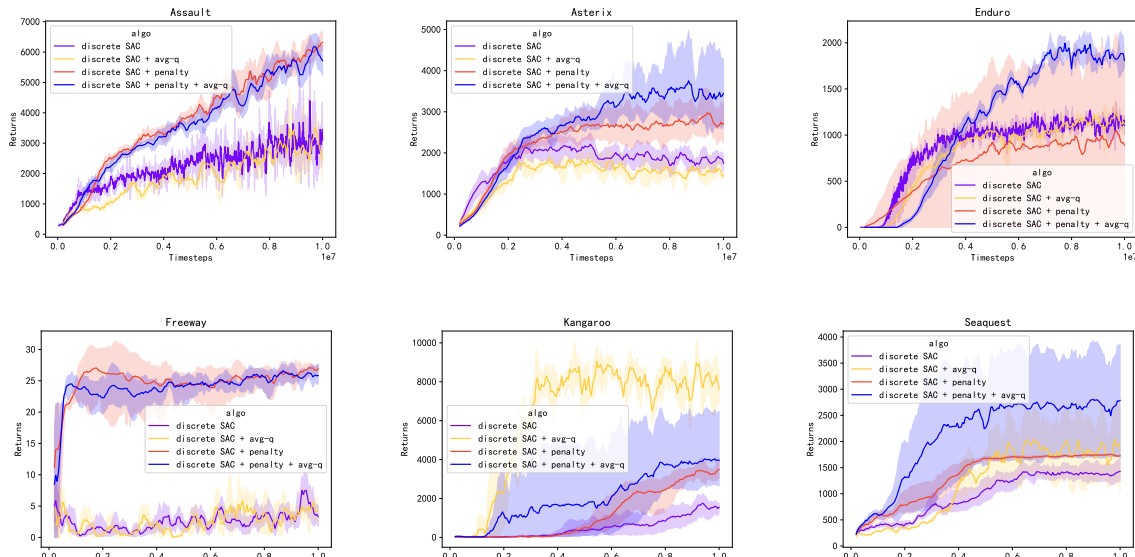

Figure 5: Scores of variant discrete SAC, which includes discrete SAC, discrete SAC with entropy-penalty, discrete SAC with double average Q learning with Q-clip,for Atari games Assault, Asterix, Enduro, Freeway, Kangaroo and Seaquest.

scores of discrete SAC and TES-SAC reported in the corresponding publication (Xu et al., 2021). Note that this is comparable as we use exactly the same benchmarks in (Christodoulou, 2019; Xu et al., 2021). When comparing our method to the discrete SAC and TES-SAC, there is a marked increase of 38% and 35.5% in mean normalized scores. And our method improves the median normalized scores by 10.7% and 9.0% while compared with discrete SAC and TES-SAC.

In order to verify the effect of longer training process, table 1 also compares discrete SAC, Rainbow, Soft-DQN our method performance on 10 million steps. Compared with discrete SAC, our method has improved the normalized scores by 68.6% and 23.3% on mean and median, respectively. Additionally, our proposed method also outperformed Rainbow, by 32.6% on mean and by 34.9% on median. Better Q-estimation and steady policy updates are responsible for the performance increase in terms of average scores. The experimental results demonstrate that benefiting from the deterministic greedy policy and entropy regularization in the

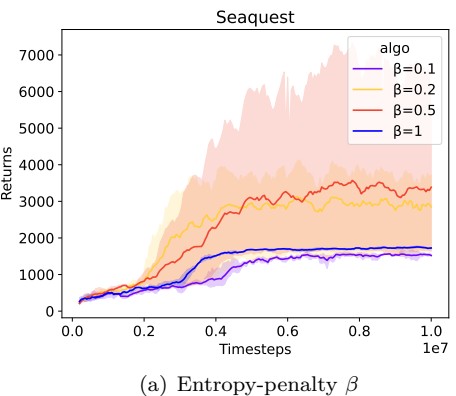 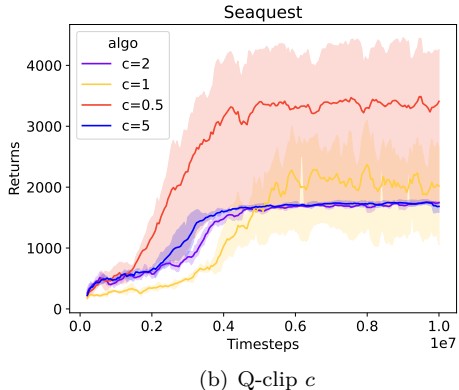

(a) Entropy-penalty $\beta$                    (b) Q-clip $c$

Figure 6: Scores on Seaquest: a) variants entropy-penalty coefficient $\beta$ with 0.1, 0.2, 0.5 and 1. b) variants Q-clip $c$ with 0.5, 1, 2 and 5.

evaluation step, Soft-DQN's performance improves rapidly in the early stages and achieves the best results at 1 million steps. However, due to the early convergence of the deterministic greedy policy, Soft-DQN's performance stagnates after 4 million steps, as seen in Fig. 9. Our method outperforms Soft-DQN in the final 10 million steps by 20.8% on average and 6.4% on median, due to the training stability brought by entropy penalty and the optimistic exploration altered by the double avg-q with q-clip.

### 6.3 Ablation Study

Fig. 5 shows the learning curves for 6 environments. Entropy-penalty (red curve) increases performance compared to the discrete SAC in each of the six environments, and even increases 2x scores in Assault. This shows that discrete SAC can obtain greater performance after removing unstable training. Except for Asterix, the alternative choice of clipped double Q-learning, which is double average Q learning with Q-clip (yellow curve), also has a certain improvement compared to the discrete SAC in 5 environments. Additional improvements can be derived when the combination of both alternative design choices is used simultaneously.

### 6.4 Hyperparameter Analysis

Our alternate design method incorporates two hyperparameters, i.e., entropy-penalty coefficient $\beta$ and Q-clip range $c$. Fig. 6 compares various entropy-penalty coefficient $\beta$ and Q-clip range $c$ values. The constraint proportion of policy change is determined by the entropy-penalty coefficient $\beta$, intuitively, an excessive penalty term will lead to policy under-optimization. We experiment with different $\beta$ in $\{0.1, 0.2, 0.5, 1\}$. We find that $\beta = 0.5$ can effectively limit entropy randomness while improving performance. Different ranges of Q value are constrained by the Q-clip range $c$, and experiments with different ranges $c$ in $\{0.5, 1, 2, 5\}$ show that 0.5 is a reasonable constraint value.

### 6.5 Qualitative Analysis

Fig. 7 shows loss surfaces of the discrete SAC and our method by using the visualization method proposed in (Li et al., 2018; Ota et al., 2021) with the loss of TD error of Q functions. According to the sharpness/flatness in these two sub-figures, our method has a nearly convex surface while discrete SAC has a more complex loss surface. When compared to the discrete SAC, the surface of our method has fewer saddle points, which further shows that our method can be more smoothly optimized during the training process.

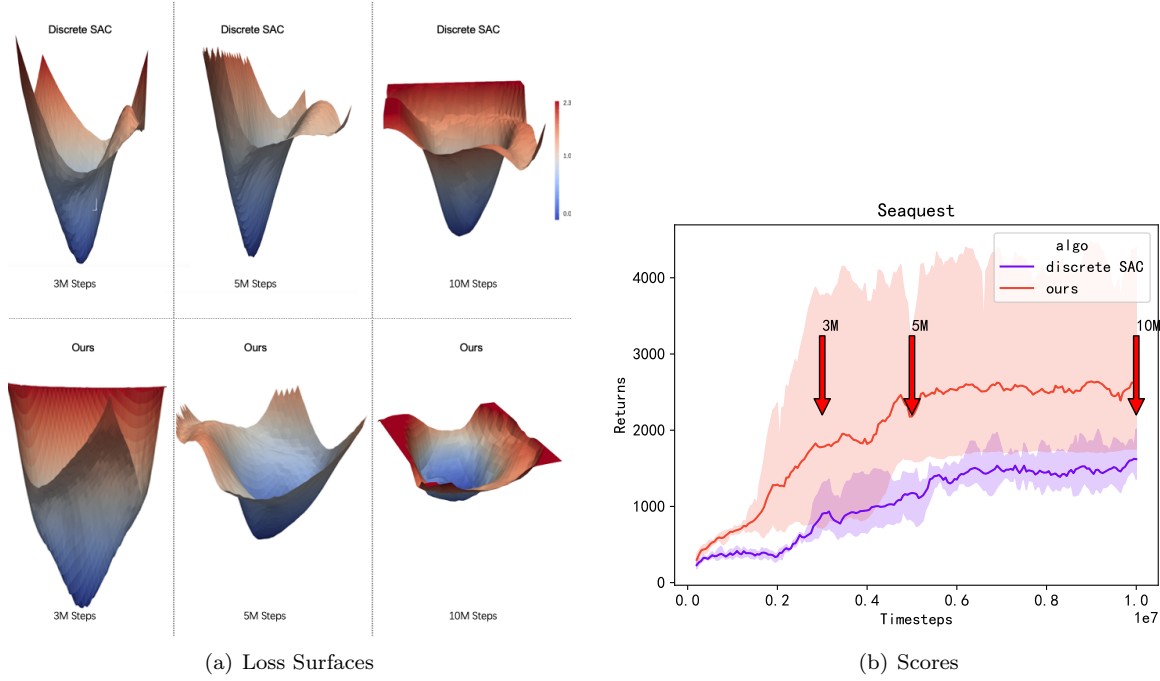

(a) Loss Surfaces

(b) Scores

Figure 7: The loss surfaces of discrete SAC and our method on Atari game Seaquest with trained weights at 3 million, 5 million and 10 million steps.

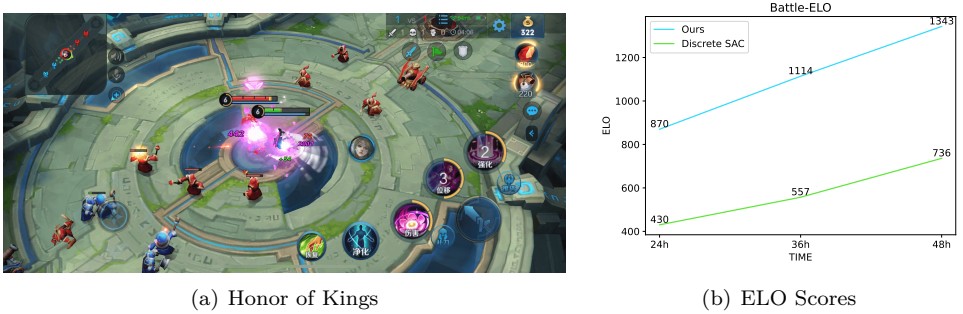

(a) Honor of Kings

(b) ELO Scores

Figure 8: a) A screenshot for Honor of Kings 1v1. b) The ELO scores compared with discrete SAC and our method, tested for three snapshots of 24, 36, and 48 hours during training.

## 7 Case Study in Honor of Kings

We further deploy our method into Honor of Kings 1v1, a commercial game in industry, to investigate the scale-up ability of our proposed SAC algorithm.

Honor of Kings is the world's most popular MOBA (Multiplayer Online Battle Arena game) and a popular testbed for RL research (Ye et al., 2020b;c;a; Chen et al., 2021a; Wei et al., 2022) The game descriptions can be found in (Ye et al., 2020c;a). In our experiments, we use the one-versus-one mode (1v1 solo), with both sides being the same hero: Diao Chan. We use the default training settings (e.g., computing resources, self-play settings, initializations, etc.) from the officially released Honor of Kings 1v1 game environment (Wei et al., 2022) (the corresponding code and tutorial are available at: `https://github.com/tencent-ailab/hok_env`). The state of the game is represented by feature vectors, as reported in (Ye et al., 2020c; Wei et al., 2022). The action space is discrete, i.e, we discretize the direction of movement and skill, same to (Ye et al., 2020c;a). The goal of the game is to destroy the opponent's turrets and base crystals while protecting

its own turrets and base crystals. The ELO rating system, which is calculated from the win rate, is used to measure the ability of two agents.

The results are shown in Fig. 8. We see that, throughout the entire training period, our method outperforms discrete SAC(Christodoulou, 2019) by a significant margin, which indicates our method's efficiency in large-scale cases.

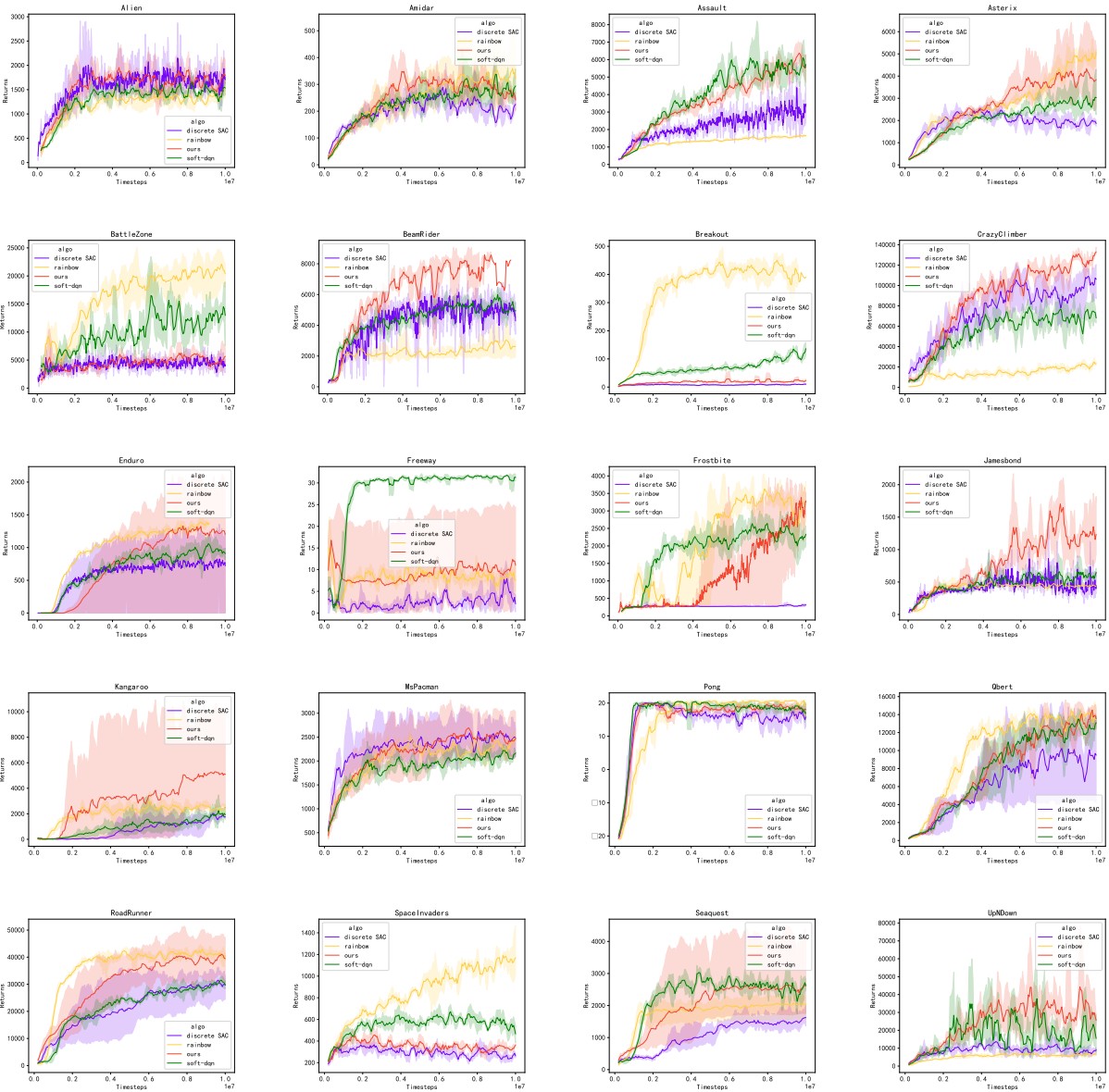

Figure 9: Learning curves for discrete SAC, Rainbow, Soft-DQN and ours, for each individual game. Every curve is smoothed with a moving average of 10 to improve readability.

## 8 Conclusions and Future Work

Many algorithmic design choices in RL are limited to the regime of the chosen benchmark tasks. Our study highlights, for the example of soft actor-critic (SAC), that widely accepted design choices in continuous

action space do not necessarily generalize to new discrete environments. We conduct failure mode analyses on Atari benchmarks, in order to understand and diagnose the implications of default design choices.

We emphasize two main insights of our discrete SAC study: 1) due to the lack of entropy constraints, unstable policy updates will further disturb the Q-value updates; 2) in addition to the overestimation bias, the underestimation bias caused by clipped double Q-learning should be taken into consideration since it results in the agent's pessimistic exploration and inefficient sample usage. We thereby propose two alternative design choices for discrete SAC, which are entropy-penalty and double-average Q-learning with Q-clip. Experiments show that our alternative design choices increase the training stability and Q-value estimation accuracy, which ultimately improves the overall performance. In addition, we also apply our method to the large-scale MOBA game Honor of Kings 1v1 to show the scalability of our optimizations.

Finally, the success obscures certain flaws, one of which is that our improved discrete SAC still performs poorly in instances involving long-term decision-making. One possible reason is that SAC can not accurately estimate the future only by rewarding the current frame. In order to accomplish long-term choices with SAC, our next study will concentrate on improving the usage of the incentive signal across the whole episode.

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

# A  Appendix

## A.1  Computation Overhead

We test the computational speed on a machine equipped with an Intel(R) Xeon(R) Platinum 8255C CPU @ 2.50GHz with 24 cores and a single Tesla T4 GPU. The unit "it/s" represents the number of steps interacting with the environment per second. Detailed data are shown in the table 3 below. The results demonstrate that our method has a 10.86% reduction(265.41->236.58) in speed compared to the vanilla discrete SAC, while maintaining the same parameter size.

Table 3: Computational speed our method and discrete SAC.

| algorithm | speed |
|---|---|
| discrete SAC | 265.41it/s |
| discrete SAC + entropy-penalty | 246.83it/s(-18.58) |
| discrete SAC + avg-q + q-clip | 250.27it/s(-15.14) |
| discrete SAC + avg-q + q-clip + entropy-penalty(ours) | 236.58it/s(-28.83) |

## A.2  Pseudo Code

---
**Algorithm 1** Discrete-SAC with entropy-penalty and double average Q-learning with Q-clip
---

Input: $\theta_1, \theta_2, \phi$        ▷ Initial parameters

Output: $\theta_1, \theta_2, \phi$        ▷ Optimized parameters

Hyperparameters: $\gamma, \beta, c, \tau$

Initialise $Q_{\theta_1} : S \to \mathbb{R}^{|A|}, Q_{\theta_2} : S \to \mathbb{R}^{|A|}, \pi_\phi : S \to [0,1]^{|A|}$        ▷ Initialise local networks

Initialise $Q_{\theta_1'} : S \to \mathbb{R}^{|A|}, Q_{\theta_2'} : S \to \mathbb{R}^{|A|}$        ▷ Initialise target networks

$\theta_1' \leftarrow \theta_1, \theta_2' \leftarrow \theta_2$        ▷ Equalise target and local network weights

$\mathcal{D} \leftarrow \emptyset$        ▷ Initialize an empty replay buffer

**for** each iteration **do**

    **for** each environment step **do**

        $a_t \sim \pi_\phi(a_t \mid s_t)$        ▷ Sample action from the policy

        $s_{t+1} \sim p(s_{t+1} \mid s_t, a_t)$        ▷ Sample transition from the environment

        $\mathbb{H}_{\pi_{old}} \sim \mathbb{E}_{a \sim \pi_\phi(\cdot \mid s_t)}[-\log \pi_\phi(a \mid s_t)]$        ▷ Calculate the entropy $\mathbb{H}_{\pi_{old}}$ of the current policy $\phi$

        $\mathcal{D} \leftarrow \mathcal{D} \cup \{(s_t, a_t, r(s_t, a_t), s_{t+1}, \mathbb{H}_{\pi_{old}})\}$        ▷ Store the transition in the replay buffer

    **end for**

    **for** each gradient step **do**

        $y \sim r(s_t, a_t) + \gamma \cdot \mathrm{avg}(Q_{\theta_1'}(s_{t+1}, \pi(s_{t+1})), Q_{\theta_2'}(s_{t+1}, \pi(s_{t+1})))$        ▷ Double average Q-value estimation

        $\mathcal{L}(\theta_i) \sim \max\left((Q_{\theta_i} - y)^2, (Q_{\theta_i'} + \mathrm{clip}(Q_{\theta_i} - Q_{\theta_i'}, -c, c)) - y)^2\right)$ for $i \in \{1, 2\}$        ▷ Clip the Q-value estimation from target critic network

        $\theta_i \leftarrow \theta_i - \lambda_Q \hat{\nabla}_{\theta_i} \mathcal{L}(\theta_i)$ for $i \in \{1, 2\}$        ▷ Update the Q-function parameters

        $\mathbb{H}_\pi \sim \mathbb{E}_{a \sim \pi_\phi(\cdot \mid s_t)}[-\log \pi_\phi(a \mid s_t)]$        ▷ Calculate the entropy $\mathbb{H}_\pi$ of policy $\phi$

        $J_\pi(\phi) \sim \mathbb{E}_{s_t \sim D}[\mathbb{E}_{a_t \sim \pi_\phi}[\alpha \log(\pi_\phi(a_t \mid s_t)) - Q_\theta(s_t, a_t)]] + \beta \cdot \frac{1}{2}(\mathbb{H}_{\pi_{old}} - \mathbb{H}_\pi)^2$

        $\phi \sim \phi - \lambda_\pi \hat{\nabla}_\phi J_\pi(\phi)$        ▷ Update policy weights

        $\alpha \sim \alpha - \lambda \hat{\nabla}_\alpha J(\alpha)$        ▷ Update temperature

        $Q_{\theta_i'} \leftarrow \tau Q_{\theta_i} + (1 - \tau) Q_{\theta_i'}$ for $i \in \{1, 2\}$        ▷ Update target network weights

    **end for**

**end for**

---

### A.3 Same Atari Environments For failure modes of DSAC

We conduct cross-validation in the same Atari environments for the failure modes, as presented in Fig.10 and Fig.11. The results show that the same three Atari games simultaneously exhibit Drastic Changes of Policy and Pessimistic Exploration problems.

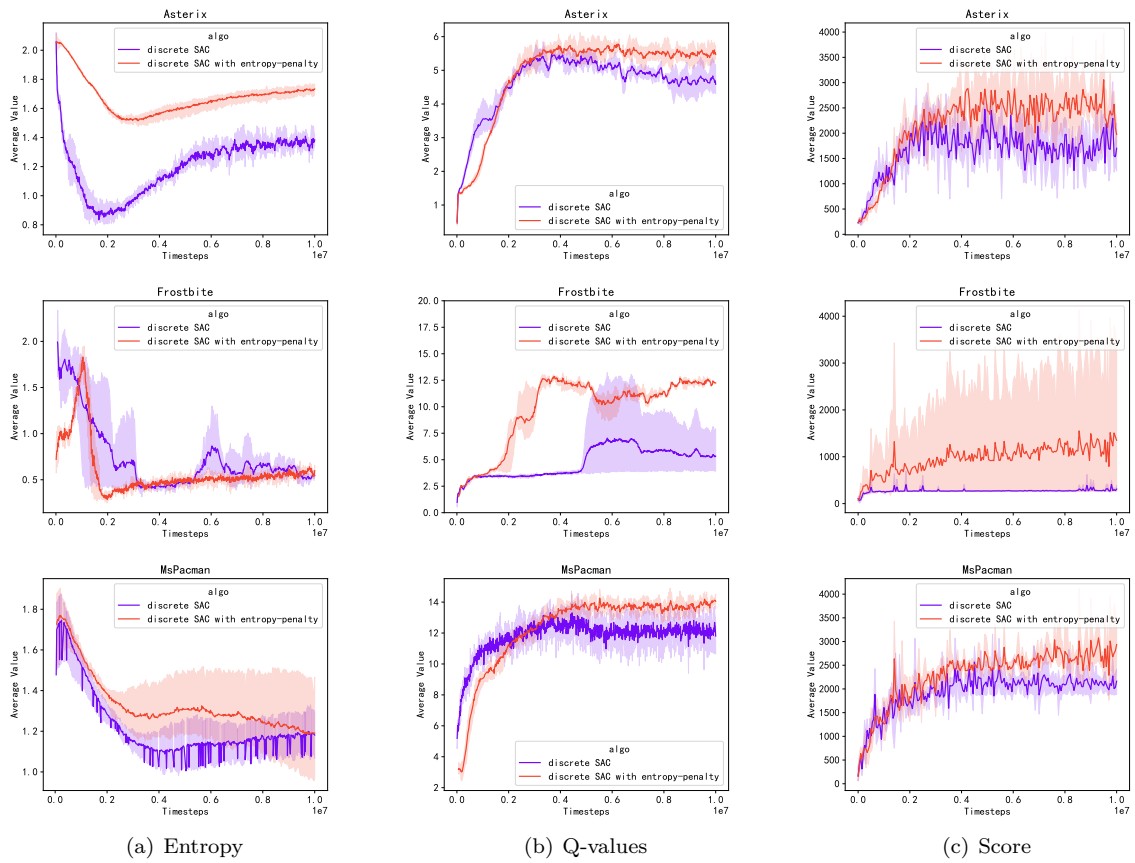

Figure 10: Measuring policy action entropy, estimation of Q-value, and score on Atari game Asterix, Frostbite , and MsPacman compared between discrete and discrete SAC with entropy-penalty over 10 million time steps

### A.4 Various Learning Rates For Drastic Changes of Policy

We introduce various learning rates for experiments on Asterix using vanilla discrete SAC in Fig.12. An excessively high learning rate leads to early convergence of entropy, while an excessively low learning rate results in insufficient optimization. The experiments show that the entropy instability issue of discrete SAC is not caused by inappropriate learning rate settings.

### A.5 Hyperparameter

### A.6 Cosine Similarity Comparison

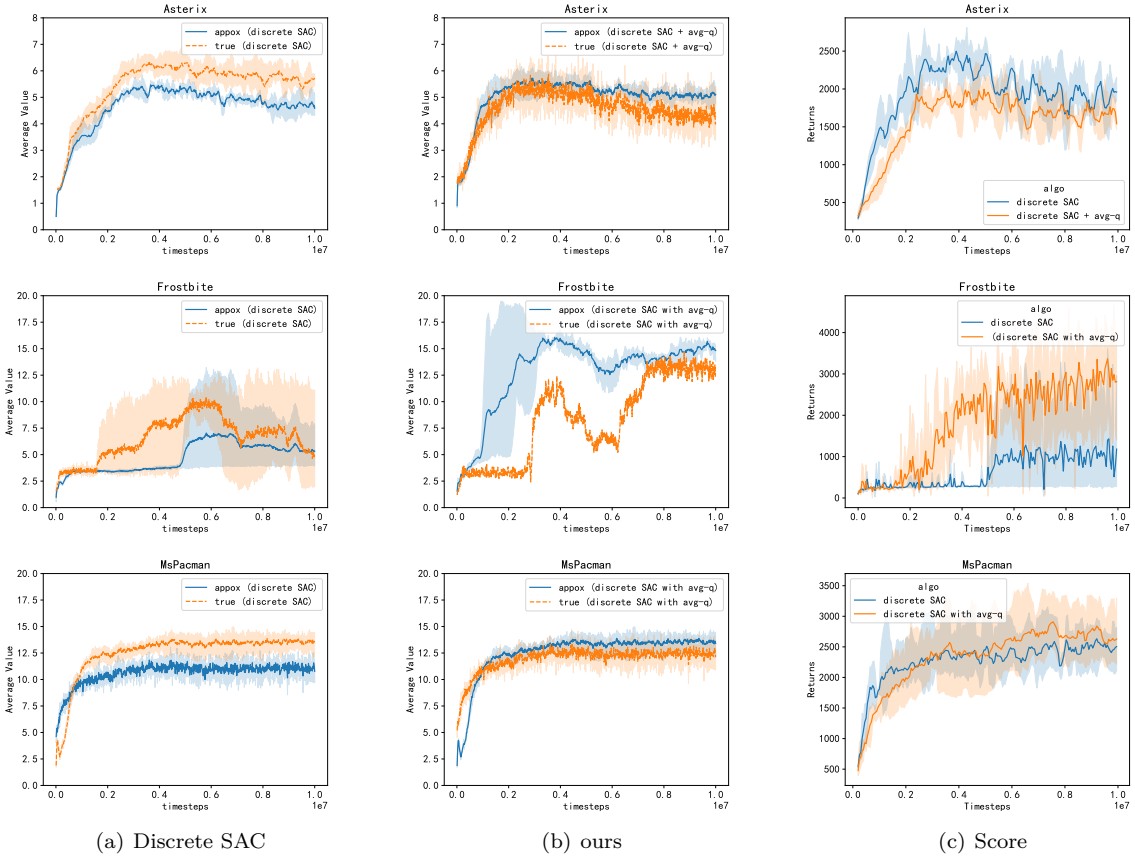

(a) Discrete SAC          (b) ours          (c) Score

Figure 11: Measuring estimation of Q-value and score on Atari Game Asterix, Frostbite and MsPacman environment compared between discrete SAC and ours (discrete SAC with double average Q-learning with Q-clip) over 10 million steps

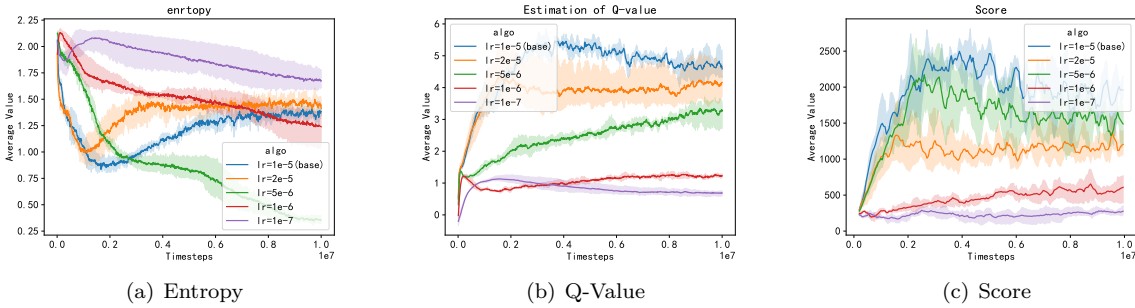

(a) Entropy          (b) Q-Value          (c) Score

Figure 12: Measuring policy action entropy, estimation of Q-value and score on Atari Game Asterix environment with discrete SAC over 10 million time steps using different learning rates.

Table 4: Hyperparameter for Discrete SAC and Ours

| Hyperparameter | Discrete SAC | Ours |
|---|---|---|
| learning rate | $10^{-5}$ | $10^{-5}$ |
| optimizer | Adam | Adam |
| mini-batch size | 64 | 64 |
| discount ($\gamma$) | 0.99 | 0.99 |
| buffer size | $10^5$ | $10^5$ |
| hidden layers | 2 | 2 |
| hidden units per layer | 512 | 512 |
| target smoothing coefficient ($\tau$) | 0.005 | 0.005 |
| Learning iterations per round | 1 | 1 |
| alpha | 0.05 | 0.05 |
| n-step | 3 | 3 |
| $\beta$ | False | 0.5 |
| $c$ | False | 0.5 |

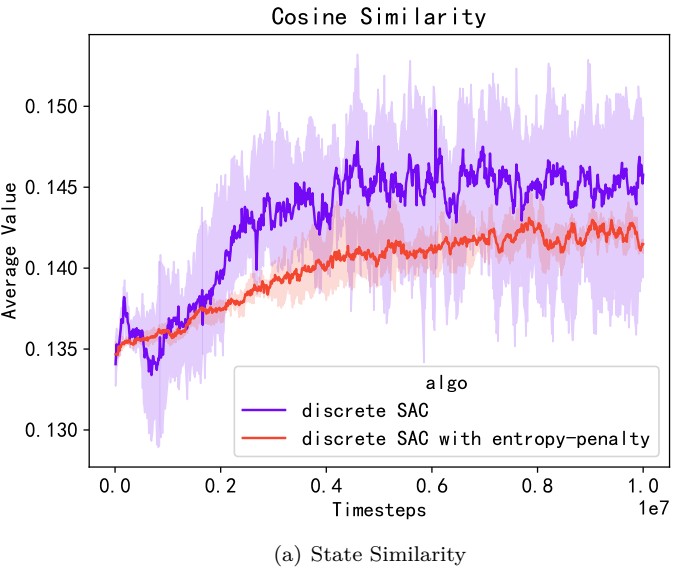

(a) State Similarity

Figure 13: Measuring cosine similarity of states on Atari game Asterix compared between discrete SAC and discrete SAC with entropy-penalty over 10 million time steps.

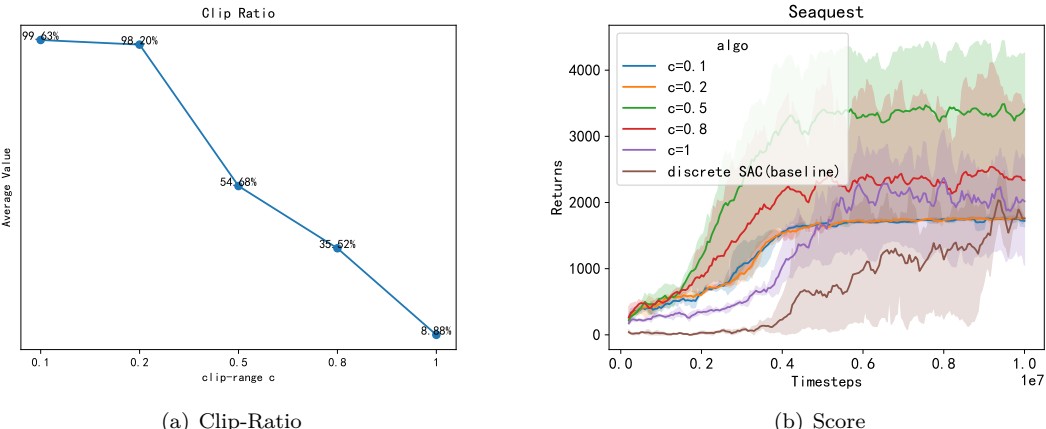

(a) Clip-Ratio

(b) Score

Figure 14: Measuring clip-ratio and score on Atari Game Seaquest environment with our method over 10 million time steps using variants Q-clip c with 0.1, 0.2, 0.5, 0.8 and 1.0 .

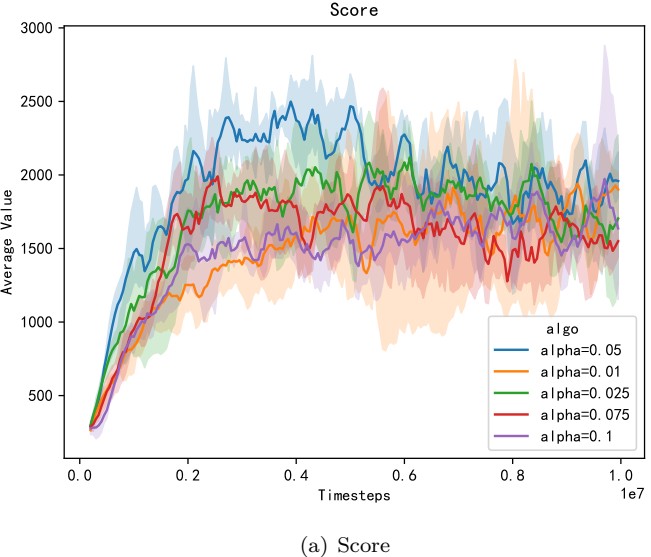

(a) Score

Figure 15: Measuring scores on Asterix by discrete SAC using variants $\alpha$ with 0.01, 0.025, 0.05, 0.075 and 0.1 over 10 million time steps.

