# OpenReview forum: "Revisiting Discrete Soft Actor-Critic"
_TMLR — Rejected by TMLR_

### Review · Reviewer_MCci · 2023-09-04

**Summary Of Contributions:**

This paper conducts an in-depth study of discrete soft actor critic. It identifies Q-value underestimation and stability issues due to fast policy updates as major issues. The paper proposes adding an entropy penalty to regularize the entropy to stay close to previous policy versions and introduces double average Q-learning with some clipping to reduce large Q updates instead of the common double Q-learning.

**Audience:**

Yes

**Claims And Evidence:**

Yes

**Requested Changes:**

Questions and requested changes:
* last paragraph of the intro: I think is a state of the art algorithm in continuous action spaces but I disagree that it is *the* SOTA. E.g. MPO and it's variants are typically stronger in my experience in off-policy settings and PPO or VMPO are generally better in terms of asymptotic performance. Please soften the language somewhat.
* why not all ATARI games?
* Can you clarify eq 14 in terms of the entropy terms? This should be part of the expectation over states, if I understand correctly?
* How do you calculate the true value in e.g. fig 2? Please address this in detail in the text.
* Please include more details about how you calculate cosine distance between state distributions in fig 1(a)
* Related works section: there's a typo in "Christodoulou at. al." and in any case the double reference is a bit confusing. Please fix.

**Strengths And Weaknesses:**

Strengths:
* nice investigation of the failure modes of discrete SAC
* extensive experimental evaluation with promising results

Weaknesses:
* the method introduces two more hyper parameters to tune and the method actually seems relatively sensitive to these choices.
* Some of the algorithmic choices seem a bit ad-hoc:
   * in the double q part why avg rather than median? or running more critics? I'd also be curious to see how distributional critics fare
   * why an entropy penalty rather than a KL constraint to the previous policy?

---

> ### Author Response · Authors · 2023-09-19
>
> Thank you for your constructive comments. We sincerely appreciate your time spent reading this paper, and we provide a point-by-point response to your comments, as follows.
>
> Q1: In the double q part why avg rather than median? or running more critics? I'd also be curious to see how distributional critics fare?
>
> A1: To provide a detailed comparison with other bias estimation approaches, we include Randomized Ensembled Double Q-learning (REDQ) [1] and Random Ensemble Mixture (REM) [2] with Vanilla Discrete SAC in our experiments. The corresponding experimental results are shown in Figure 4.
>
> The results demonstrate that even though REDQ has less estimation bias, it still suffers from underestimation bias, leading to suboptimal performance due to pessimistic exploration. We also find that, although REM addresses the underestimation issue, the overestimation bias of REM significantly exceeds that of our proposed method, resulting in a rapid decline in performance at 8 million steps.
> We find that our proposed method, despite introducing some overestimation bias, can enhance exploration and improve the policy, which is in line with the principle of "optimism in the face of uncertainty" [3].
>
> Q2: Why an entropy penalty rather than a KL constraint to the previous policy?
>
> A2: We introduce a dynamic KL penalty (as in PPO [4]) on newer and older policies as a new baseline. As shown in Figure 3, compared to Vanilla Discrete SAC, the KL penalty can mitigate the drastic policy changes, preventing training collapse. However, we find that the effectiveness of the KL penalty in reducing drastic changes is inferior to the entropy penalty(please refer to the entropy curve in Figure 3(a)). The final Q-value and score (reward from the test environment) are lower than those with the entropy penalty, with a difference of 12% and 23%, respectively.
>
> By adding regularization in entropy space instead of policy space, our method, the entropy penalty, can mitigate the drastic changes of policy entropy while maintaining the inherent exploratory ability of SAC.
>
> Q3: The method introduces two more hyperparameters to tune and the method actually seems relatively sensitive to these choices.
>
> A3: In our experiments, we conduct a grid search over various hyperparameter combinations on 5 Atari games and perform a detailed comparative analysis of the experimental results in Figure 6. We find that the parameter $\beta$ works well in the range of 0.2-0.5, and the parameter $c$ works stably in the range of 0.5-1 -- this shows that our method is not very sensitive to these hyperparameters. Therefore, we apply $\beta$=0.5 and $c$=0.5 on all 20 Atari environments. The experiments demonstrate that the introduced hyperparameters exhibit good robustness in various Atari environments.
>
> Q4: How do you calculate the true value in e.g. fig 2? Please address this in detail in the text.
>
> A4: The true value in Figure 2 is represented by the unbiased Monte Carlo return: $V_t=R_{t+1}+\gamma R_{t+2}+\gamma^2 R_{t+3}+\cdots+\gamma^{T-t-1} R_T$. We use Monte Carlo return to measure the estimation bias of the Q function.
>
> Q5: Please include more details about how you calculate cosine distance between state distributions in fig 1(a).
>
> A5:  At each training iteration, we calculate the cosine distance between state distributions as below. First, we collect trajectories ($\tau_{old}=s_1,s_2,...$, $\tau_{new}=s_{1}^{\prime},s_{2}^{\prime},...$) using $\pi_{old}$ and $\pi_{new}$ respectively. Second, we calculate the cosine distance between these trajectories as the following: $ \frac{1}{N} \sum_{t=1}^N cosine distance(s_t, s_{t}^{\prime})$.
>
> Q6: Why not all ATARI games?
>
> A6:  Following our baseline methods (Soft Actor-Critic for Discrete Action Settings [5]), we choose the same 20 Atari games for fair comparison. As described in [5], the selected games vary significantly and were chosen a priori and so we believe the results on these 20 games are a good estimate for relative performance on the whole Atari suite of 49 games.
>
> Q7：Can you clarify eq 14 in terms of the entropy terms? This should be part of the expectation over states, if I understand correctly?
>
> A7: Following your suggestion, we rewrite the entropy term calculation in Equation 14 in the form of a part of the expectation over states in our revision, which is $$J_{\pi}(\phi)=E_{s_{t} \sim D}[E_{a_{t} \sim \pi_{\phi}}[\alpha \log (\pi_{\phi}(a_{t} \mid s_{t}))-Q_{\theta}(s_{t}, a_{t})]] + \beta \cdot \frac{1}{2} E_{s_{t} \sim D} ([ E_{a_{t} \sim \pi_{\phi_{old}}} [-  \log(\pi_{\phi_{old}})  ] - E_{a_{t} \sim \pi_{\phi}} [ - \log(\pi_{\phi}) ])^{2}$$

---

> ### Author Response · Authors · 2023-09-19
>
> Q8: Other minor issues:
> * Related works section: there's a typo in "Christodoulou at. al." and in any case the double reference is a bit confusing. Please fix.
> * Last paragraph of the intro: I think is a state of the art algorithm in continuous action spaces but I disagree that it is the SOTA. E.g. MPO and it's variants are typically stronger in my experience in off-policy settings and PPO or VMPO are generally better in terms of asymptotic performance. Please soften the language somewhat.
>
> A8: We thank reviewer for pointing out these issues and we have revised them in our revision.
>
> The following are the references：
>
> [1] Chen et al. Randomized ensembled double q-learning: Learning fast without a model. In International Conference on Learning Representations, 2021b.
>
> [2] Agarwal at al. An optimistic perspective on offline reinforcement learning. In Proceedings of the 37th International Conference on Machine Learning, ICML 2020.
>
> [3] Ciosek at al. Better exploration with optimistic actor critic. Advances in Neural Information Processing Systems, 32, 2019.
>
> [4] Schulman at al. Trust region policy optimization. Proceedings of the 32nd International Conference on Machine Learning 2015.
>
> [5]Petros Christodoulou. Soft actor-critic for discrete action settings. arXiv preprint arXiv:1910.07207, 2019.

---

### Review · Reviewer_kXKw · 2023-09-04

**Summary Of Contributions:**

The paper proposes a new SAC variant that works in the discrete setting. 2 techniques are proposed: a) a penalty that will make the current entropy of policy to be closer to the entropy of an older policy. and b) use average of Q estimates instead of min as in clipped double Q.
The paper compares to Rainbow and the original Discrete SAC and show the proposed variant can outperform them on a set of Atari tasks. The paper also shows that the proposed method is stronger than Discrete SAC on Honor of Kings 1v1.

**Audience:**

Yes

**Claims And Evidence:**

No

**Requested Changes:**

- Related work section needs work
- Please make labels and legend larger in the figures to make them more readable.
- Center the figure captions to make them look nicer
- Can be good to have more analysis to support your claims and have more comparison to newer related work, recent SOTA methods, and other baselines with existing techniques.

**Strengths And Weaknesses:**

**Strengths**
- The overall writing is clear.
- The empirical results can be interesting.

**Weaknesses**
- Some arguments are not well supported: Top of page 3, I don't quite see why both "Q-value and policy fall into local optimum", the avearge Q value does not really show whether it's good or bad. And the policy's performance has continued to improve until it hits a plateau. It is unclear how the paper comes to the local optimum conclusion?
- Figure 2 and Figure 4: it seems the proposed method is not really fixing the estimation bias issue, I find it hard to believe the proposed method can be applied to a general setting and still work very well.
- The loss landscape in RL can be constantly changing, in Figure 6 is this only for this particular environment at a particular stage during training, or is it a consistent observation?

**Related work**
- Since this paper studies the bias issue, it is important to consider some more recent works that study this issue, for example, Maxmin q-learning: Controlling the estimation bias of q-learning (Lan et al.), which study over and under-estimation issue in discrete setting; Controlling overestimation bias with truncated mixture of continuous distributional quantile critics (Kuznetsov et al.), Randomized ensembled double q-learning: Learning fast without a model (Chen et al.) which propose finer control of bias for SAC in continuous setting. And even these papers are out 2, 3 years ago. The paper needs a better discussion of related work and on how your proposed method is different from, or better than these developments in the literature.
- Rainbow is also quite old, the comparison in the paper shows the proposed method can beat discrete SAC and rainbow, but can it compete with more recent SOTA methods?

**Novelty**
- The missing related work also means the novelty of this work might be limited.

**More ablations**
- How does the proposed method compare to existing (sometimes naive) techniques? For example, can we simply use policy delay as used in TD3? Or use KL penalty on newer and older policies? There is no evidence in the paper showing why the proposed method is necessary and improves over an existing technique.

Some other questions:
- Figure 5 why there is a massive variance in many environments?
- It might be interesting to provide more details on the Honor of Kings experiments.
- How much more computation overhead does your method introduce? Currently it's unclear. Can have a computation table in appendix.

---

> ### Author Response · Authors · 2023-09-19
>
> Thank you for your constructive comments. We sincerely appreciate your time spent reading this paper, and we provide a point-by-point response to your comments as follows.
>
>
> Q1: More comparison to newer related work for bias issues.
>
> A1:  Regarding Q-value estimation, our proposed method, Double Average Q-learning with Q-clip, is designed to address the issue of insufficient exploration caused by underestimation rather than just reducing estimation bias.
>
> To provide a detailed comparison with other bias estimation approaches, we include Randomized Ensembled Double Q-learning (REDQ) [1] and Random Ensemble Mixture (REM) [2] with Vanilla Discrete SAC in our experiments. The corresponding experimental results are shown in Figure 4. The results demonstrate that even though REDQ has less estimation bias, it still suffers from underestimation bias, leading to suboptimal performance due to pessimistic exploration.  We also find that, although REM addresses the underestimation issue, the overestimation bias of REM significantly exceeds that of our proposed method, resulting in a rapid decline in performance at 8 million steps.
>
> We find that our proposed method, despite introducing some overestimation bias, indeed enhances exploration and improves the policy, which is in line with the principle of “optimism in the face of uncertainty” [3].
>
>
> Q2: How does the proposed method compare to existing (sometimes naive) techniques? For example, can we simply use policy delay as used in TD3? Or use KL penalty on newer and older policies? There is no evidence in the paper showing why the proposed method is necessary and improves over an existing technique.
>
> A2: In comparison to existing techniques regarding drastic changes of policy, since Vanilla Discrete SAC has already integrated policy delay (used in TD3), we introduce a dynamic KL penalty (as in PPO [4]) on newer and older policies as a new baseline.
>
> As shown in Figure 3, compared to Vanilla Discrete SAC, the KL penalty can mitigate the drastic changes of policy, preventing training collapse. However, we find that the effectiveness of the KL penalty in reducing drastic changes is inferior to the entropy penalty (please refer to the entropy curve in Figure 3(a)). The final Q-value and score (reward from the test environment) are lower than those with the entropy penalty, with a difference of 12% and 23%, respectively.
>
> By adding regularization in the entropy space instead of the policy space, our method, the entropy penalty, can mitigate the drastic changes of policy entropy while maintaining the inherent exploratory ability of SAC.
>
>
> Q3：Some arguments are not well supported: Top of page 3, I don't quite see why both "Q-value and policy fall into local optimum", the average Q value does not really show whether it's good or bad. And the policy's performance has continued to improve until it hits a plateau. It is unclear how the paper comes to the local optimum conclusion?
>
> A3：In Figure 1, the score represents the average return, which indicates the performance of the policy. We observe that the score no longer increases after 3 million steps, which is consistent with the trend of the Q-value curve after 3 million steps. This suggests that the policy exploration ceases to grow as the Q-value no longer increases, leading to a local optimum.
>
> Q4: The loss landscape in RL can be constantly changing, in Figure 6 is this only for this particular environment at a particular stage during training, or is it a consistent observation?
>
> A4：We plot the loss landscape during training at 3 million, 5 million, and 10 million steps in Figure 7. We observe that the loss landscapes at different training phases show a consistent trend with what we described in our previous paper.
>
> Q5: In Figure 5, why there is a massive variance in many environments?
>
> A5: In our experiments, we follow the common practice (as in PPO [4]), using three random seeds for each algorithm and running each method for 10 million steps. We rerun our experiments using more random seeds. The results are shown in Figure 5 of the updated paper after the revision. We find that with more random seeds, the variance is reduced, and our method still outperforms the baselines, which is consistent with our pervious observation.

---

> ### Author Response · Authors · 2023-09-19
>
> Q6:How much more computation overhead does your method introduce? Currently it's unclear. Can have a computation table in appendix.
>
> A6：We test the computational speed on a machine equipped with an Intel(R) Xeon(R) Platinum 8255C CPU @ 2.50GHz with 24 cores and a single Tesla T4 GPU. The unit "it/s" represents the number of steps interacting with the environment per second. Detailed data are shown in the table below. The results demonstrate that our method has a 10.86% reduction(265.41->236.58) in speed compared to the vanilla discrete SAC, while maintaining the same parameter size.
> |  algorithm   | speed  |
> |  ----  | ----  |
> |                      discrete SAC                     				 |      265.41it/s     		|
> |             discrete SAC + entropy-penalty           			 |  246.83it/s(-18.58) 	|
> |             discrete SAC + avg-q + q-clip             			 |  250.27it/s(-15.14) 	|
> | discrete SAC + avg-q + q-clip + entropy-penalty(ours)  |  236.58it/s(-28.83) 	|
> |                  discrete SAC + REDQ                  			 | 106.96it/s(-158.45)	|
> |                   discrete SAC + REM                  			 |  89.82it/s(-175.59) 	|
>
> Q7: Need to discuss the difference between the proposed method and other related work.
>
> A7: Maxmin Q-learning [5] controls estimation bias by minimizing over the full ensemble in the target. MME extends max-min operation to the entropy framework to adapt to SAC. REM [6] ensembles Q-value estimations with a random convex combination to enhance generalization in the offline setting. REDQ [1] reduces the estimation bias by minimizing a random subset of Q-functions. AEQ [7] adjusts the estimation bias by using the mean of Q-functions minus their standard deviation. Compared to these methods that employ ensemble multiple Q-estimators, our approach focuses on reducing underestimation bias for double Q-estimators to enhance exploration. In addition, the introduced computational overhead of our method is much less than that of the methods above (as shown in the table of Q6A6).
>
> Following your suggestion, we include the discussion in our new revision.
>
> Q8: Rainbow is also quite old, the comparison in the paper shows the proposed method can beat discrete SAC and rainbow, but can it compete with more recent SOTA methods?
>
> A8: Our proposed method aims to address the "Drastic Changes of Policy" and Pessimistic Exploration issues caused by the underestimation of Q-value in the discrete domain of SAC. We believe that it is orthogonal to other state-of-the-art methods, and it would be an interesting future research direction to combine them and investigate if there is further performance improvement.
>
> Q9: It might be interesting to provide more details on the Honor of Kings experiments.
>
> A9: To adapt our method from Atari to Honor of Kings, we make the following adjustments:
>
> 1.We use n-step TD-learning (n=20) to accelerate the reward propagation and enhance the long-term decision-making of both discrete SAC and our method since Honor of Kings is an environment with sparse reward.
>
> 2.Our experiments find that Double Average Q-learning with Q-clip, due to its optimistic estimation, provides better exploration in the larger exploration space of the Honor of Kings environment, making it easier to explore efficient policies and achieve more coherent skills in the early stages of training (at 24 hours).
>
> 3.We discover that the entropy-penalty significantly enhances the stability of the self-play training process in the King's environment. In contrast, the vanilla discrete SAC in self-play training tends to fall into a mode collapse where agents focus on killing minions rather than attacking enemy heroes.
>
> We provide a GIF in the supplementary materials, where the blue agent represents our method and the red agent represents discrete SAC. The GIF demonstrates that our agent releases more coherent combos and has stronger killing capabilities.
>
> Other minor issues:
>
> Q10: Please make labels and legend larger in the figures to make them more readable and center the figure captions to make them look nice.
>
> A10: Thanks for your suggestions. We have revised these issues in our revision. We have reorganized the labels, legends, and figure captions to improve readability.

---

> ### Author Response · Authors · 2023-09-19
>
> The following are the references：
>
> [1] Chen et al. Randomized ensembled double q-learning: Learning fast without a model. In International Conference on Learning Representations, 2021b.
>
> [2] Agarwal at al. An optimistic perspective on offline reinforcement learning. In Proceedings of the 37th International Conference on Machine Learning, ICML 2020.
>
> [3] Ciosek at al. Better exploration with optimistic actor critic. Advances in Neural Information Processing Systems, 32, 2019.
>
> [4] Schulman at al. Trust region policy optimization. Proceedings of the 32nd International Conference on Machine Learning 2015.
>
> [5] Lan at al. Maxmin q-learning: Controlling the estimation bias of q-learning. In 8th International Conference on Learning Representations, ICLR 2020.
>
> [6] Seungyul Han and Youngchul Sung. A max-min entropy framework for reinforcement learning.  Advances in Neural Information Processing Systems. NeurIPS 2021.
>
> [7] Gong et al. Adaptive estimation q-learning with uncertainty and familiarity. In Proceedings of the Thirty-Second International Joint Conference on Artificial Intelligence, IJCAI 2023, pp. 3750–3758. ijcai.org, 2023.

---

### Review · Reviewer_NPAN · 2023-09-20

**Summary Of Contributions:**

This paper focuses on discrete-SAC. SAC is a popular entropy-regularized off-policy RL algorithm for continuous domains, here adapted here for discrete ones.

The work exhibits two weaknesses of the base algorithm, and propose a solution  for both of them. First, they show that the entropy of the policy changes very rapidly. To alleviate that, they add an entropy regularization term (between to successive entropies) to the policy objective. Then, they show that discrete SAC suffers from pessimistic exploration, due to the use of double q-learning. Here, they propose to replace the minimum in double q-learning by a mean, and they add an additional clipping to the q-values.

These improvements are then tested on several domains, including the standard Atari one, where the show good performance compared to standard baselines.

**Audience:**

Yes

**Broader Impact Concerns:**

This is algorithmic advances tested on synthetic environments. It  does not require a Broader Impact Statement .

**Claims And Evidence:**

No

**Requested Changes:**

I think this paper could be improved with some changes.

**Discrete version of SAC** (weakness 1)

1. Regarding weakness 1, I think the introduction could be slightly reworded. In particular, I don't think the statements around the lines of "SAC [...] cannot be straight-forwardly applied to discrete domains since it relies on the reparameterization of Gaussian policies to sample actions" are totally accurate (see the weakness section for details).

2. Still regarding weakness 1, I think this method should compare to a soft q-learning with only q-value parametriztion. This would be a more natural baseline.

**Motivation for entropy penalty** (weakness 2)

3. On the motivation for the entropy penalty:
 - I think there could be some clarifications around weaknesses 2.1 and 2.3
- In 2.2, I suggested an experiment that could be valuable to the motivation for this penalty.

**Experiments** (weakness 3)

4. On weakness 3, I agree it is not always realistic to run the Atari benchmark for 200M steps. If this is not possible, I would suggest toning down the claims of outperforming SOTA, or comparing to a baseline more optimized for sample efficiency.

**Strengths And Weaknesses:**

# Strengths

**Clarity**

The paper is easy to read an understand. In particular, the main idea -- identifying flaws in D-SAC and providing solution -- is clearly laid out, and well organized.

**Relevance**

SAC is a popular algorithm (for good reason), so these lines of work are relevant to the whole RL community.


# Weaknesses

I think this work has 3 main weaknesses.

## 1. Adaptation of SAC to discrete domains does not need policy parametrization

One of the weaknesses of this work is that SAC-discrete is not the natural adaptation of SAC to discrete actions. Indeed, at its essence, SAC is an entropy-regularized value iteration based algorithm. In this  framework, the greedy policy becomes a softmax of the q-value (instead of a max without the entropy), which is highlighted as softmax is the solution to Eq. 7.  The Bellman update consists in computing a logsumexp of the q-value, which can be retrieved from Eq. 6. So, in a discrete action settings:
 -  we can compute the greedy step with softmax
 -  we can compute the evaluation step with a logsumexp,
and therefore we do not need a policy network, but only a q-value parametrization.

Thus, SAC is already an adaptation of a discrete algorithm to continuous actions, and the discrete equivalent of SAC is a double q-learning algorithm with only a q-value parametrization (for example double DQN), with entropy regularization (using a logsumexp instead of a max in the Bellman update). An implementation and evaluation of this method on the full Atari suite was done in [2], but the idea dates back from before, this idea is for example detailed in section 3 of [1].

That said, it does not mean that using methods such as discrete SAC is a bad idea, but the baseline comparison should be a this q-learning method. In  this work, we can see that there can be an interest in having a policy loss – adding arbitrary regularizations there, like the entropy one – but we do not know how it impacts performance in the first place. There is no evidence that using an approximation is better than just computing the softmax.

[1] Haarnoja, Tuomas, et al. "Reinforcement learning with deep energy-based policies." International conference on machine learning. PMLR, 2017.

[2] Vieillard, Nino, Olivier Pietquin, and Matthieu Geist. "Munchausen reinforcement learning." Advances in Neural Information Processing Systems 33, 2020.

## 2. Motivation for the additional entropy regularization

The entropy penalty introduced in Eq. 14 makes sense, but  think it lacks motivation, on two axes: why we need it, and why this design choice.

**2.1: some unclear statements in section 4.1**

I do not really understand the paragraph following the section 4.1 title. Could you comment what you mean by "However, due to the existence of entropy term in the soft bellman error, the policy update iteration (Eq. 8) is strongly coupled with the Q-learning iteration (Eq. 5)." They are coupled in any actor-critic algorithm by construction, so I don't understand why it is an argument here.

 **2.2: example of section 4.1**

I am not convinced by the example used in Section 4.1. SAC controls the entropy through an adaptive parameter with a loss that targets a specific value (Eq. 10). In Figure 1.b, the behavior we observe  is a typical result of such a control, maybe with a learning rate set too high. A way too check this would be to have the experiment of figure 1.b with different values for the learning rate of the alpha loss.

**2.3: design choice**

Last point is that it lacks a discussion on why this design choice was made, compared to a more standard KL regularization. I think it is not clear from the paper what entropy penalty achieves that could not be done by a KL penalty, this could be discussed. Additionally, with a KL penalty, we know what  the optimal solution of the policy optimization is, do you have an idea of the solution for the optimization problem in Eq. 14 ?


## 3. Atari experiments

I am not sure it makes a lot of sense to compare a method to Rainbow 10M, an algorithm that was specifically optimized for the 200M setup. Its performance at 10M steps is not very informative. There have been many improvements to these methods to make them more sample efficient, which would make for a better SOTA baseline in this setup (a latest example is BBF [3]).

[3] Schwarzer, Max, et al. "Bigger, Better, Faster: Human-level Atari with human-level efficiency." International Conference on Machine Learning. PMLR, 2023.

# 4. Minor remarks
- in the intro, (Ciosek et al., 2019; Pan et al., 2020) cite seems to be misplaced
- Formatting: following eq. 10, notations on target entropy are not well formatted.
- Eq. 14; the entropy notation is not super clear, and I think it should be inside of the expectation over states ? *[Edit after first revision: fixed]*
-  Atari setting: instead of the setting used by (Mnih et al. , 2013), I would suggest following the recommendation of [4] (sticky actions, no end-of-episode signal).

[4] Machado, Marlos C., et al. "Revisiting the arcade learning environment: Evaluation protocols and open problems for general agents." Journal of Artificial Intelligence Research 61 (2018): 523-562.

---

> ### Author Response · Authors · 2023-09-28
>
> Thank you for your constructive comments. We sincerely appreciate your time spent reading this paper, and we provide a point-by-point response to your comments as follows.
>
> Q1：Discrete version of SAC
>
> Q1.1 Regarding discrete version of SAC, this method should compare to a soft q-learning with only q-value parametriztion. This would be a more natural baseline.
>
> A1.1: Following your suggestion, we implement **Soft-DQN from [1] as a new baseline for comparison**. The summary of the results is shown in the following table. The raw scores and curves are shown in Table 1, Table 2, and Figure 9 in the revised version:
> | |Discrete SAC(1M)  | TES-SAC(1M) | Soft-DQN(1M)   | Ours(1M)   | Rainbow(10M) | Discrete SAC(10M)       | Soft-DQN(10M) | Ours(10M) |
> | ---- |----|----|----|----|----|----|----|----|
> |Mean  |    0.5% | 3.0% |  41.7% | 38.5%  | 187.4 % | 151.4% | 199.2% | 220.0%|
> |Median | 0.4% | 2.1% | 20.0 | 11.1%  | 79.2 % |90.8% | 107.7%  | 114.1 |
>
> The experimental results demonstrate that benefiting from the deterministic greedy policy and entropy regularization in the evaluation step, Soft-DQN's performance improves rapidly in the early stages and  achieves the best results at 1 million steps.
> However, due to the early convergence of the deterministic greedy policy, Soft-DQN's performance stagnates after 4 million steps, as seen in Figure 10. Our method outperforms Soft-DQN in the final 10 million steps by 20.8% on average and 6.4% on median, owing to the training stability brought by the entropy penalty and the optimistic exploration altered by the double avg-q with q-clip. From the experimental results, we speculate that utilizing a policy in conjunction with a stable Q-value approximation, rather than solely deriving the policy from the softmax of Q-value, can prevent the policy from rapidly falling into local optima.
>
> Q1.2：Regarding weakness 1, i don't think the statements around the lines of 'SAC [...] cannot be straight-forwardly applied to discrete domains since it relies on the reparameterization of Gaussian policies to sample actions' are totally accurate (see the weakness section for details).
>
> A1.2:  Thanks for pointing out this issue. We rephrase this statement in the revised version (paragraph 2 in the introduction) as "However, while SAC solves problems with continuous action space, it cannot be straight-forwardly applied to discrete domains since it relies on the reparameterization of Gaussian policies to sample actions, in which the action in discrete domains is categorical. Soft-DQN [1] provides a simple way to discretize SAC by adopting the maximum-entropy RL to DQN. However, Soft-DQN utilizes only a Q-value parametrization to bypass the policy parameterization."
>
> Soft-DQN is another discretization method for SAC, so we also include Soft-DQN as one of our baselines. Empirically, we find that Soft-DQN performs better than vanilla discrete SAC. This paper aims at improving discrete SAC, and we discover that with our optimization, discrete SAC can outperform Soft-DQN. It would be interesting to explore how to improve Soft-DQN in the future.
>
> [1] Vieillard, Nino, Olivier Pietquin, and Matthieu Geist. "Munchausen reinforcement learning." Advances in Neural Information Processing Systems 33, 2020.

---

> ### Author Response · Authors · 2023-09-28
>
> Q2: Motivation for entropy penalty
>
> Q2.1: I do not really understand the paragraph following the section 4.1 title. Could you comment what you mean by "However, due to the existence of entropy term in the soft bellman error, the policy update iteration (Eq. 8) is strongly coupled with the Q-learning iteration (Eq. 5)." They are coupled in any actor-critic algorithm by construction, so I don't understand why it is an argument here.
>
> A2.1: One of our motivations in this paper is to address the issue of training instability of discrete SAC within the framework of maximum entropy RL, and we find that this issue stems from the drastic changes in entropy during training, which is closely related to the coupled update between Q-value and policy. Therefore, we propose an entropy penalty to mitigate this problem. In other actor-critic algorithms (such as DDPG [2] and PPO [3]), the update of Q value and policy is coupled as well. However, the effect of the coupled update varies across different algorithms. For example, in DDPG, the coupled update leads to overestimation, while in PPO, it results in a trade-off between variance and bias.
>
>
> Q2.2: Answer the design choice of entropy penalty.
>
> A2.2: The KL penalty constrains the distribution distance between the old policy and the new policy, preventing the new policy from deviating too far from the old one. Different from the KL penalty, the entropy penalty for discrete SAC is motivated by the **desire to promote exploration and maintain diversity and stability  in the learned policy.** By penalizing the drastic policy changes, the algorithm encourages the agent to choose diverse actions, which in turn can help the agent to better explore the environment, maintain stability, and avoid getting stuck in local optima.
>
> As comparison, we introduce a dynamic KL penalty (as in PPO [3]) on newer and older policies as a new baseline. As shown in Figure 3, compared to vanilla discrete SAC, the KL penalty can mitigate the drastic policy changes, preventing training collapse. However, we find that the effectiveness of the KL penalty in reducing drastic changes is inferior to the entropy penalty(refer to the entropy curve in Figure 3(a)). The final Q-value and score (reward from the test environment) are lower than those with the entropy penalty, with a difference of 12% and 23%, respectively.
>
> By adding regularization in entropy space instead of policy space, our method can mitigate the drastic policy changes while maintaining the inherent exploratory ability of SAC.
>
>
> Q2.3: Regarding weakness2.2 , is an inappropriate learning rate setting causing the issue in Figure 1.b?
>
> A2.3: We introduce **various learning rates (base*2, base/2, base/10, base/100) for experiments** on Asterix using vanilla discrete SAC In appendix Fig 12. An excessively high learning rate (lr=2e-5) leads to early convergence of entropy, while a shallow learning rate (lr<=5e-6) results in insufficient optimization. The experiments show that the entropy instability issue of discrete SAC is not caused by inappropriate learning rate settings.
>
> [2] Lillicrap, Timothy P , et al. Continuous control with deep reinforcement learning. International Conference on Learning Representations, ICLR 2016
> [3] Schulman et al. Trust region policy optimization. Proceedings of the 32nd International Conference on Machine Learning 2015.
>
>
> Q3: On weakness 3, I agree it is not always realistic to run the Atari benchmark for 200M steps. If this is not possible, I would suggest toning down the claims of outperforming SOTA, or comparing to a baseline more optimized for sample efficiency.
>
> A3: Thanks for your suggestion. We revise the paper's wording to tone down the claims of outperforming SOTA in the revision.
>
> Q4: Atari setting: instead of the setting used by (Mnih et al. , 2013), I would suggest following the recommendation of [4] (sticky actions, no end-of-episode signal).
>
> A4: Thank you for your suggestion. Our previous experiments aim for a fair comparison between Rainbow and discrete SAC, so we adopt the same environment settings from [5].  Our propose method yields similar conclusions when extended to the Honor of Kings 1v1 environment, indicating that our approach is robust across different environment settings.
>
> [4] Machado, Marlos C , et al. "Revisiting the arcade learning environment: Evaluation protocols and open problems for general agents." Journal of Artificial Intelligence Research 61 (2018): 523-562.
>
> [5] Mnih et al. Playing atari with deep reinforcement learning. CoRR, abs/1312.5602, 2013. URL http://arxiv.org/abs/1312.5602.
>
> Q5: Other minor issues:
> * in the intro, (Ciosek et al., 2019; Pan et al., 2020) cite seems to be misplaced
> * Formatting: following eq. 10, notations on target entropy are not well formatted.
>
> A5: We thank reviewer for pointing out these issues and we have revised them in our revision.

---

> > ### Comment · Reviewer_NPAN · 2023-10-05
> > **Clarification on Q2.3**
> >
> > Thank you for your detailed answers, it clarifies most of my questions.
> >
> > I would like to ask a clarification about your answer to Q 2.3:
> >
> > When I mentioned the learning rate, I was talking about the one use for the loss that deals with the automatic enropy adjustment (Eq 10), not about the learning rate of the policy or the value loss. Is this actually the one you tested for ?
> >
> > If it is, then I am quite surprised by the results in Figure 12 : since SAC with a constant alpha works reasonably well, how do you explain that just changing the speed of alpha variation has such an impact on performance ?

---

> > > ### Author Response · Authors · 2023-10-11
> > >
> > > Q6: When I mentioned the learning rate, I was talking about the one use for the loss that deals with the automatic enropy adjustment (Eq 10), not about the learning rate of the policy or the value loss. Is this actually the one you tested for ? If it is, then I am quite surprised by the results in Figure 12 : since SAC with a constant alpha works reasonably well, how do you explain that just changing the speed of alpha variation has such an impact on performance ?
> > >
> > > A6: Firstly, in Fig. 12, we test various learning rates for policy and value loss rather than the automatic enropy adjustment. Secondly, for the baseline implementation of discrete-SAC, we use Tianshou [1]. In Tianshou, the official implementation includes auto-alpha in continuous SAC and fixed-alpha in discrete SAC. We find that Tianshou's implementation [2] (fixed-alpha at 0.05) performs better than the original paper by Christodoulou [3]. Therefore, we use the default hyperparameters in Tianshou, which includes a constant alpha, and the hyperparameters are consistent across all 20 games. Since we adopt the fixed-alpha mechanism, we do not provide separate experimental results for various learning rates, only for automatic entropy adjustment.
> > >
> > > Additionally, as shown in Figure 15, we tune the alpha for discrete SAC and find that Tianshou's default setting (alpha=0.05) already achieves optimal performance. Consequently, for all 20 Atari environments, we utilize alpha=0.05 as the default parameter.
> > >
> > > [1] [GitHub - thu-ml/tianshou: An elegant PyTorch deep reinforcement learning library.
> > > ](https://github.com/thu-ml/tianshou)
> > >
> > > [2] https://github.com/thu-ml/tianshou/tree/master/examples/atari#sac-single-run
> > >
> > > [3] Petros Christodoulou. Soft actor-critic for discrete action settings. arXiv preprint arXiv:1910.07207, 2019.

---

### Review · Reviewer_vWe7 · 2023-09-20

**Summary Of Contributions:**

The paper conducts an examination of the SAC algorithm in the discrete setting. This work finds two failure-modes due to which the vanilla discrete SAC algorithm shows a poor performance empirically.

First, they claim that the un-regularized entropy term may lead to abrupt changes in the state distribution. This can lead to the Q-learning target being unstable decreasing the stability of the policy. To address this, the authors propose a entropy penalty term that keeps the entropy of the learned policy is close to the behavior policy.

Secondly, they demonstrate the min operation in the Double-Q learning mechanism may actually cause an underestimation of Q-values. The authors propose a fix for this by replacing the min operation with an average operation.

Finally, the authors compares their modified discrete SAC algorithm with the vanilla discrete SAC and Rainbow using 20 ATARI environments and Honor of Kings

**Audience:**

Yes

**Claims And Evidence:**

No

**Requested Changes:**

1. The issues raised in Weakness 1,2 and 3 should be addressed.
2. A more detailed discussion of the hyper-parameters would be useful. It seems like there are a couple of very specific hyperparameters that makes the method work. The other two hyper-parameter shows a performance which seems worse than the discrete SAC and Rainbow.
3. Would adding TEC-SAC in Fig 3 set a stronger baseline than the vanilla discrete SAC? It may also help address weakness 1.
4. Typo in the caption of Fig 1: Atati -> Atari

**Strengths And Weaknesses:**

# Strength
1. The empirical results show that the modified discrete SAC outperforms the discrete SAC and Rainbow for a number of ATARI games.
2. The work is relevant to the RL community with SAC being a popular choice of algorithm.
3. The paper is easy to read and understand. The experiments are aligned with the motivation of the paper.

# Weakness
1. The primary weakness in this work is that the claim about unregularized maximum entropy term leading to unstable Q-values is not really demonstrated. In Fig 1. we can see that even if there are sharp changes in entropy, there are no corresponding sharp changes in the Q-value. If anything, the estimated Q-value is in line with the returns show in Fig 1(d). This indicates that there are no extrapolation errors due to sudden changes in the state distribution.

2. It is likely that the performance gains from the modifications comes with better exploration in the early stages of the training. This can be seen in Fig 3(a) where the entropy for modified SAC stays high between 1M and 3M timesteps. It is well-known that the SAC is sensitive to hyperparameters ((Haarnoja et al., 2018b, Xu et al., 2021). This modification introduces another hyperparameter which needs to be tuned well for the modification to work (Fig 6).

3. The underestimation claim for the double-Q learning probably would not generalize to most environments. In cases where $\epsilon_1$ > $\epsilon_2$ > 0 or $\epsilon_1$ > 0 > $\epsilon_2$, the average operation may lead to overestimation. Furthermore, replacing min with avg does not show any improvement in Fig 5.

4. The choice of environment changes with every figure. Is there a reason why a the ATARI environments are not the same for every figure?

---

> ### Author Response · Authors · 2023-09-28
>
> Thank you for your constructive comments. We sincerely appreciate your time spent reading this paper, and we provide a point-by-point response to your comments as follows.
>
> Q1: Regarding weakness 1, “The primary weakness in this work is that the claim about unregularized maximum entropy term leading to unstable Q-values is not really demonstrated. In Fig 1. we can see that even if there are sharp changes in entropy, there are no corresponding sharp changes in the Q-value. If anything, the estimated Q-value is in line with the returns show in Fig 1(d). This indicates that there are no extrapolation errors due to sudden changes in the state distribution.”
>
> A1:  In Fig. 1(b), from 0 to the first 4 million steps, it is observed that the entropy of the vanilla discrete SAC **decreases rapidly from 2.0 to 0.8**, which we define as a sharp change in policy. Additionally, the Q-value **rapidly increases to 5 during the initial 4 million steps; however, after 4 million steps, the Q-value decreases from 5 to 4.5**, as illustrated in Fig. 1(c), which is consistent with the trend of the score in Fig 1(d).
>
> This suggests that due to the **rapid convergence of the policy** (entropy decreases sharply) in the early stages of training, the policy performance ceases to improve as the Q-value no longer increases and even declines after convergence, leading to a local optimum.
>
> In contrast, as shown in Fig 3, our method exhibits a smooth decrease in entropy during the first 4 million steps, and the Q-value stabilizes after rising to 5. The final score demonstrates that our performance is better than discrete SAC.
>
>
> Q2: This modification introduces another hyperparameter which needs to be tuned well for the modification to work (Fig 6).
>
> A2: In our experiments, we conduct a grid search over various hyperparameter combinations on 5 Atari games and perform a detailed comparative analysis of the experimental results in Figure 6. We find that the parameter $\beta$ works well in the range of 0.2-0.5, and the parameter $c$ works stably in the range of 0.5-1. This shows that our method is not very sensitive to these hyperparameters. Therefore, we apply $\beta$=0.5 and $c$=0.5 on all 20 Atari environments. The experiments demonstrate that the introduced hyperparameters exhibit robustness across various Atari environments.
>
>
> Q3：The underestimation claim for the double-Q learning probably would not generalize to most environments. In cases where $\epsilon_1 > \epsilon_2 > 0$ or $\epsilon_1 > 0 > \epsilon_2$ , the average operation may lead to overestimation. Furthermore, replacing min with avg does not show any improvement in Fig 5.
>
> A3:  In response to the overestimation issue in cases where $\epsilon_1 > \epsilon_2 > 0$ or $\epsilon_1 > 0 > \epsilon_2$, our **clip-q** mechanism in Eq. 17 can protect the critic estimation from exceeding the estimation range c of the target critic, thus preventing the Q-values from collapsing due to excessive overestimation.
>
> The performance of double avg-q with q-clip is inferior to double min-q on Asterix because discrete SAC itself has severe **training instability issues** in Asterix (refer to Fig 3). Relying solely on double avg-q with q-clip solves the underestimation problem, but the issue of drastic policy changes still remains. When we introduce both double avg-q with q-clip and entropy penalty, we address both issues of drastic policy changes, and underestimation bias and our method outperforms discrete SAC. We find that this can can be generalized to all the tested environments, as shown in Fig 5.
>
>
> Q4: A more detailed discussion of the hyper-parameters？
>
> A4: We introduce two parameters: entropy penalty coefficient $\beta$ and Q-clip range $c$, with their reasonable ranges being 0.2-0.5 and 0.5-1, respectively. The results of hyper-parameters are shown in Fig.5.
>
> In policy update iterations, a too-small $\beta$ (such as $\beta<=0.1$) causes the entropy penalty to degrade to vanilla discrete SAC. At the same time, a too-large $\beta$ (such as $\beta >=1$) makes the policy optimization direction more focused on entropy stability rather than policy optimization.
>
> In the evaluation step, when we use a too-large clip range (such as $c >= 1$ ), the overestimation error introduced by double avg-q will accumulate in each update step. In contrast, with a too-small clip range (such as $c <= 0.5$ ), the Q-value update will become conservative which restricts the improvement of both Q-value and policy .

---

> > ### Comment · Reviewer_vWe7 · 2023-10-05
> > **Response to the authors' comments**
> >
> > Thank you for your response. Please find follow up questions and comments below.
> >
> > Re A1:In section 4.1, there is a claim that "the abrupt change of state distribution could lead to policy instability". Could you comment on how this claim is verified in Figure 1. I understand that the policy saturates at a suboptimal local minima but does it show a link between abrupt change in state distribution and the suboptimal policy? Moreover, if the proposed method of entropy regularization addresses this, would it be possible to add a plot similar to Fig 1(a) in Fig (3) that shows a comparison of the state distribution changes between Discrete SAC and Discrete SAC with entropy regularization?
> >
> > Re A2: Fig (6) shows that if $\beta$ changes from 0.2 to 0.1, there is a substantial drop in performance. It seems that the parameter c is even more sensitive and a change from 0.5 to 1 based on your response shows a substantial drop in performance. I find it difficult to agree that the method is not very sensitive to hyper-parameters
> >
> > Additional question Q8: The results for the discrete SAC are taken from the paper( Christodoulou, 2019). The authors in the paper mention that the results come from the untuned algorithm. In this paper, a tuned modification of discrete SAC is compared with an untuned discrete SAC. Did you try tuning the hyper-parameters (especially $\alpha$) of the unmodified discrete SAC to set a stronger baseline?

---

> ### Author Response · Authors · 2023-09-28
>
> Q5: Would adding TEC-SAC in Fig 3 set a stronger baseline than the vanilla discrete SAC? It may also help address weakness 1.
>
> A5: We choose to compare with TES-SAC because it is similar to our method in terms of adjusting the entropy. Since their code is not publicly available and we fail to reproduce the results in their paper, we cannot compare the differences between our method and theirs regarding entropy, Q-value, and other metrics to provide more analysis. Therefore, in our experiments, we choose to directly compare our results with their reported scores in the original TES-SAC paper.
>
> Q6: The choice of environment changes with every figure. Is there a reason why a the ATARI environments are not the same for every figure?
>
> A6: Although we use different Atari environments in Section 4 to demonstrate various issues, these problems are, in fact, universally present. To show this, we provide more case studies, as presented in Appendix Fig. 10 and Fig. 11. The results show that the Atari games (Asterix, Frostbite, and MsPacman) that appear in different figures  simultaneously exhibit drastic policy changes and pessimistic exploration issues.
>
> Q7: Other minor issues:
> * Typo in the caption of Fig 1: Atati -> Atari
>
> A7: We thank reviewer for pointing out this issue and we have revised it in our revision.

---

> ### Author Response · Authors · 2023-10-11
>
> Q8: The results for the discrete SAC are taken from the paper( Christodoulou, 2019). The authors in the paper mention that the results come from the untuned algorithm. In this paper, a tuned modification of discrete SAC is compared with an untuned discrete SAC. Did you try tuning the hyper-parameters (especially $\alpha$) of the unmodified discrete SAC to set a stronger baseline?
>
> A8: For the baseline implementation of discrete-SAC, we use **Tianshou [1] (which is tuned for parameters and incorporates n-step learning)**. We find that Tianshou's implementation performs better than the original paper by Christodoulou. We use the default hyperparameters in Tianshou for both discrete SAC and our method when comparing the final results at 10 million steps.
>
> Additionally, the final parameters tuned by Tianshou are shown below:
>
>
> | Hyperparameter | Discrete SAC(paper) |Discrete SAC(tianshou) | Ours |
> | ---- |----|----|----|
> | optimizer |Adam | Adam | Adam |
> | batch size | 64 | 64 | 64 |
> | discount rate($\gamma$) | 0.99 | 0.99| 0.99 |
> | hidden layers | 2| 2 | 2 |
> |hidden units per layer |512 | 512 | 512|
> | target smoothing coefficient ($\tau$) |0.005 |  0.005 | 0.005|
> | Learning iterations per round |1 | 1 | 1 |
> | ---- |----|----|----|
> | learning rate | $ 3 * e^{-4}$  | $ 1 * e^{-5}$ | $ 1 * e^{-5}$ |
> | buffer size |$1 * e^{6}$| **$1 * e^5$** | $1 * e^5$ |
> | n-step | **False** | **3** | 3 |
> | Auto-alpha | **True** | **False** | False |
> | alpha | **False** |**0.05** | 0.05 |
> |Entropy target| **0.98 * (-log (1 / \|A\|))** | **False** | False |
> |$\beta$| False | False | **0.5** |
> |$c$| False | False | **0.5** |
>
> [1] [GitHub - thu-ml/tianshou: An elegant PyTorch deep reinforcement learning library.](https://github.com/thu-ml/tianshou)
>
> Q9: In section 4.1, there is a claim that "the abrupt change of state distribution could lead to policy instability". Could you comment on how this claim is verified in Figure 1.
>
> A9: We measure the cosine distance between state distributions induced by adjacent policies (i.e., $\pi_k$ and $\pi_{k+10}$). The range of value changes reflects the shifts in state similarity throughout the training process.
>
> As shown in Fig. 13, the cosine distance value of discrete SAC rapidly increases from 0.135 to 0.145 between 2 million and 4 million steps, indicating an abrupt change in state distributions. In contrast, under the constraint of entropy penalty, the cosine distance value is lower than that of discrete SAC and **gradually rises to 0.14**, demonstrating that our method can effectively constrain drastic policy changes, which facilitates policy training.
>
> The instability issue in discrete SAC originates from the lack of constraint when there is an abrupt change in state distribution, and our proposed entropy penalty can mitigate this issue.
>
>
> Q10: Fig (6) shows that if  $ \beta$ changes from 0.2 to 0.1, there is a substantial drop in performance. It seems that the parameter $c$ is even more sensitive and a change from 0.5 to 1 based on your response shows a substantial drop in performance. I find it difficult to agree that the method is not very sensitive to hyper-parameters.
>
> A10： Regarding the issue of sensitivity to hyper-parameters in the proposed method, we have seriously considered it in our experiments.
>
> On the one hand, the performance remains stable when the hyperparameter $\beta$ lies within the range of 0.2-0.5. On the other hand, the hyperparameter $c$ exhibits sensitivity, specifically as its performance is significantly better when $c$ is set to 0.5 compared to other choices. We conjecture that this is because the Q-value is more sensitive. To further illustrate this issue, we provide an experimental comparison of the clip ratio (indicating the proportion of the clip mechanism in effect) and the final performance with different $c$ values. As shown in Figure 14, the clip ratio decreases as c increases. When $c=0.1$  or $c=0.2$, the small clip range makes the policy conservative, as 99% of Q-value samples are clipped. As $c$ increases to 0.5, the clip ratio drops to 54.68%, effectively mitigating overestimation while maintaining optimistic exploration. When c=1, the clip ratio is only 8.88%, which limits the ability to constrain Q. The clip-ratio curve explains why the hyperparameter c performs well around 0.5. Our experiments show that when *$c=0.5$*, it performs well in most environments, including 20 Atari games and Honor of Kings, making it a general option.
>
> Additionally, we acknowledge that the choice of hyperparameters has a significant impact on the performance of our method, which is a challenge faced by most machine learning methods.  Consequently, we plan to explore reducing our method's sensitivity to hyperparameters in future work. We will use more adaptive methods or other techniques to address this issue.
>
> Once again, thank you for your review and valuable suggestions. We will consider your suggestions and further improve our method in future work.

---

### Author Response · Authors · 2024-11-27

Our major revision has been accepted by TMLR, and it was published on the 23rd of November, 2024. Please find the link to our updated paper (submission 3135) here: https://openreview.net/forum?id=EUF2R6VBeU.

---

### Decision · Action_Editor_ufQx · 2023-11-12

**Recommendation:** Reject

**Comment:**

Given the authors are improving over a widely-used algorithm, in order to get other people use this method, it may be good for the authors to carefully update the claims about the "two issues", by either doing more controlled experiments to support the "issue". Another option is to simply remove some of the heuristic claims and focus on reporting empirical results (e.g. comparison of entropy penalty, KL, and maybe some others, and show that entropy penalty is the clear best).

A side issue raised in the reviews is that the concept of "exploration" could be very vague and be perceived as different things by different readers. To me it feels this concept may not be central to the issues / proposed methods anyways. I suggest reducing its use in the and focus on more objective terminologies ("abrupt change of state distributions" or "Q value underestimation" are good examples).

**Audience:**

Given the popularity of SAC in the RL community, the results in this paper would be of interest to the audience.

**Claims And Evidence:**

This paper identifies two issues with the Discrete Soft Actor-Critic (Discrete SAC) algorithm: policy instability, and Q value underestimation, and proposes fixes to the algorithm that addresses both issues.

While the reviewers are generally positive about the empirical performance of the algorithm, whether the two identified issues are well-supported were questioned by several reviewer. In particular, the "abrupt state distribution change" issue (Section 4.1) seems lacking a more rigorous evidence (e.g. by adding some more controlled experiments). The proposed solution to this is a new "entropy penalty" term that aligns the entropy of the new (updated) and the old policy. As the authors indicated in the rebuttals, such a choice is important and other natural choices such as KL regularization does not work as well. In my opinion, the importance of this particular choice makes the "abrupt distribution change" an insufficient explanation, and makes it even more necessary to have a rigorous understanding.

**Resubmission Of Major Revision:**

The authors may consider submitting a major revision at a later time.